# Finding the gap: neuromorphic motion-vision in dense environments

Thorben Schoepe [1,2,3,4] ✉, Ella Janotte[5], Moritz B. Milde[6], Olivier J. N. Bertrand[7], Martin Egelhaaf[7] & Elisabetta Chicca[2,3,4]

Animals have evolved mechanisms to travel safely and efficiently within different habitats. On a journey in dense terrains animals avoid collisions and cross narrow passages while controlling an overall course. Multiple hypotheses target how animals solve challenges faced during such travel. Here we show that a single mechanism enables safe and efficient travel. We developed a robot inspired by insects. It has remarkable capabilities to travel in dense terrain, avoiding collisions, crossing gaps and selecting safe passages. These capabilities are accomplished by a neuromorphic network steering the robot toward regions of low apparent motion. Our system leverages knowledge about vision processing and obstacle avoidance in insects. Our results demonstrate how insects might safely travel through diverse habitats. We anticipate our system to be a working hypothesis to study insects' travels in dense terrains. Furthermore, it illustrates that we can design novel hardware systems by understanding the underlying mechanisms driving behaviour.

Animals travel in a variety of habitats, from bare landscapes to highly cluttered terrains, such as forests or grass and flower meadows. One critical aspect of flying animals, such as insects and birds, is avoiding obstacles to prevent wing damage and to enable fast locomotion. Obstacle avoidance requires maintaining a safe distance from surrounding objects, identifying crossable gaps between objects, as well as rapidly decelerating if the flight corridor gets too narrow or performing evasive manoeuvres around obstacles. The computational mechanisms underlying the highly virtuosic flight manoeuvres of insects and birds are being unravelled largely in artificial settings, such as flight tunnels with a variety of obstacle constellations[1–4] or gaps in flight barriers with variable clearance[5–11], which can be manipulated in a targeted way by the experimenters.

The primary information for flight control and obstacle avoidance of insects and birds is provided by optic flow (OF). Translational OF is the apparent motion of the surroundings on the animal's retina when the animal translates within its environment[12,13]. Since OF depends on both the velocity of locomotion as well as the distance to objects in the environment, it is directly related to the time of an impending collision[14,15]. To date, multiple mechanisms have been proposed to explain the further processing of OF to ensure collision-free flight. Strategies closely inspired by the behaviour and neuronal substrate of insects involve balancing the OF experienced in different regions of the eye (e.g. left and right hemisphere)[16,17], integrating the motion across the entire visual field[18], relying on the contrast of the OF between the foreground and background[7], using optimised spatial sensitivity[19], or learning an association between active vision (oscillation of the agent) and the apparent size of objects[20]. The use of artificial agents allows us to rigorously test the functionality of the underlying neuronal processes in controlled environments. Especially with respect to insects, it is impressive that they safely accomplish their virtuosic navigational feats, including obstacle avoidance, with a brain no bigger than a pinhead. This suggests the underlying neural processes to be extremely efficient and computationally parsimonious

[1]Peter Grünberg Institut 15, Forschungszentrum Jülich, Aachen, Germany. [2]Faculty of Technology and Cognitive Interaction Technology Center of Excellence (CITEC), Bielefeld University, Bielefeld, Germany. [3]Bio-Inspired Circuits and Systems (BICS) Lab. Zernike Institute for Advanced Materials (Zernike Inst Adv Mat), University of Groningen, Groningen, Netherlands. [4]CogniGron (Groningen Cognitive Systems and Materials Center), University of Groningen, Groningen, Netherlands. [5]Event Driven Perception for Robotics, Italian Institute of Technology, iCub facility, Genoa, Italy. [6]International Centre for Neuromorphic Systems, MARCS Institute, Western Sydney University, Penrith, Australia. [7]Neurobiology, Faculty of Biology, Bielefeld University, Bielefeld, Germany. ✉e-mail: t.schoepe@fz-juelich.de

and the underlying computational mechanisms to be potentially interesting for implementation in resource-saving artificial autonomous agents. Therefore, one central objective of the present study is to explain important aspects of the obstacle avoidance behaviour of insects by modelling them and discussing these bio-inspired computational principles with regard to technical application scenarios.

We used a closed-loop neuromorphic approach to investigate whether asynchronous processing of OF enables collision-free navigation in a multitude of environments. Neuromorphic hardware processes and transmits information in an event-based manner. For example, a neuromorphic camera (or event-based camera) sends events asynchronously only if a relative change in luminance over time is observed in a given pixel[21–25]. Neuromorphic processors, on the other hand, integrate incoming events using spiking neurons. These artificial units communicate asynchronously with one another using action potentials, or so-called spikes. The sampling scheme incorporated in neuromorphic systems is referred to as Lebesgue sampling[26], which equips these systems with high-temporal resolution and fast processing strategies. Since the obstacle avoidance machinery of insects is thought to be based on OF information (driven by the visual sense) and the initial visual processing steps are spatially distributed and asynchronous, a neuromorphic camera is suitable for studying mechanisms underlying collision-free navigation inspired by the insects' neuronal machinery and behaviour in various terrains. In search of a possible mechanism for collision-free navigation, we constrained our neuromorphic model agent by known properties of the fruit fly's visual motion pathway, which has been described at a neuronal level[27]. The apparent motion of surrounding objects is processed in several columnar organised layers of neurons in the optic lobe. The T4 and T5 neurons, which exist in each retinotopic column in the second neuronal layer of the visual motion pathway, are thought to be at the output of elementary local motion processing in flies[28,29]. Large neurons spatiotemporally integrate the responses of the elementary motion detectors and project to descending neurons conveying OF information to structures in the centre of the brain (the central complex and fan-shaped body) and to the motor control centres. At these downstream neuropils, integration of different information streams takes place[27,30], and the locomotory movements are orchestrated. Therefore, we mimicked the neuronal response of T4/T5 cells and spatiotemporally integrated them. These signals are then processed in a neural network, providing direction information for locomotion control to avoid collisions. We assessed our neuromorphic model agent's obstacle avoidance and gap-finding capabilities based on closed-loop simulations and real-world experiments in various environments. Our test environments were similar to those in which the corresponding biological evidence for obstacle avoidance of flying insects was obtained (see above). Our agent stays away from walls in a box, centres in corridors, crosses gaps and meanders around obstacles in cluttered environments. These results illustrate that neuromorphic principles that replicate an entire action-perception loop can provide a useful heuristic tool for understanding the complex behaviour of biological systems. Besides, the computationally parsimonious neuromorphic principles implemented in our agent could also be applied in autonomous vehicles.

## Results

The task for our neuromorphic and bio-inspired agent is to travel safely in different environments. Insects accomplish this task notably by extracting optic flow (OF) retinotopically and integrating this information across their visual field[31]. Our agent's inner computation replicates this processing. The Spiking Neural Network (SNN) model tested in closed-loop within different environments consists of two main components, namely a retinotopical map of insect-inspired motion detectors, i.e. spiking Elementary Motion Detectors (sEMDs)[32], and an inverse soft Winner-Take-All (WTA) network, as previously

suggested in open-loop for collision avoidance[32,33] (see Fig. 1f and Supplementary Fig. 3). The former extracts OF, which, during translational motion, is inversely proportional to the agent's relative distance to objects in the environment. The latter searches for a region of low apparent motion, hence an obstacle-free direction (see Fig. 1a–e). The advantage of this search using the inverse soft WTA is the response flexibility, which may increase the robustness in contrast to proposed alternatives, such as balancing the OF or responding to the average or maximum OF. After the detection of such a path in the environment, the agent executes a turn towards the new movement course. We characterised the network in two steps. First, we evaluated the sEMD's response and discussed similarities to its biological counterpart, i.e. T4/T5 neurons, which are thought to be at the output of elementary motion processing in fruit flies[28,29]. Second, to further demonstrate the real-world applicability of sEMD-based gap finding in an SNN, we performed closed-loop experiments. We simulated an agent seeing the world through an event-based camera in the Neurorobotics physical simulation platform[34]. The camera output was processed by the SNN, resulting in a steering command. We tested the performance of this simulated agent with the same parameters in all reported experimental conditions hereafter. These experimental conditions were inspired by experiments with flying insects. Additionally, we evaluated the performance of the algorithm during a corridor-centering task in a real-world environment.

### Spiking elementary motion detector

The sEMD represents an event-driven adaptation for neuromorphic sensory-processing systems of the well-established correlation-based elementary motion detector[35]. To evaluate the response of the sEMD in the Nest simulator[36], we compared the normalised velocity tuning curves of its ON-pathway (only events generated by light increments) to the corresponding normalised tuning curve of *Drosophila*'s T4 and T5 neurons[37]. Both velocity tuning curves are determined in response to square-wave gratings with 100% contrast and a wavelength of 20° moving at a range of constant velocities (with temporal frequencies from 0.1 to 10 Hz). The sensory data used in this paper were recorded with a DVS PAER128 event-based camera using the slow dvs128 bias setting provided in jAER viewer (for more details, see Supplementary Notes 1 and Supplementary Notes 2). Similar to the tuning curve of *Drosophila*'s T4 and T5 neurons, sEMD preferred direction exhibits a bell-shaped velocity tuning curve (see comparison Fig. 2a and b), which has the maximum response (mean population activity) at 5 Hz (100 deg s⁻¹). The null direction response is much lower than that to the preferred direction.

The sEMD model, which is composed of two macro pixels (2 × 2 pixels) of an event-based camera, two Spatio-Temporal Correlation (SPTC) neurons to remove uncorrelated noise and one Time Difference Encoder (TDE) (see Fig. 1d), exhibits a drop in its output response when the temporal frequency exceeds 5 Hz as can be seen in Fig. 2a. This drop is, however, not explained by the TDE's transfer function (see Subsection "Spiking elementary motion detector"). We would expect the response to saturate at high temporal frequencies since the TDE produces interspike intervals and firing rates inversely proportional to the time difference between the two inputs of the TDE. Rather than being a consequence of the motion detector model itself, we hypothesise that the drop in response is a consequence of the spatio-temporal band-pass filtering installed by the SPTC layer. While low temporal frequencies lead to unambiguous spatiotemporally correlated and causal SPTC spikes from adjacent neurons, high temporal frequencies lead to anti-correlated and non-causal spikes. Thus, the TDE can no longer (spatially) match the spikes unambiguously, which results in a bell-shaped velocity tuning curve of the preferred direction response.

A similar bell-shaped velocity tuning curve can be observed in *Drosophila*'s T4 cells (see Fig. 2b)[14,28,37]. While *Drosophila*'s velocity

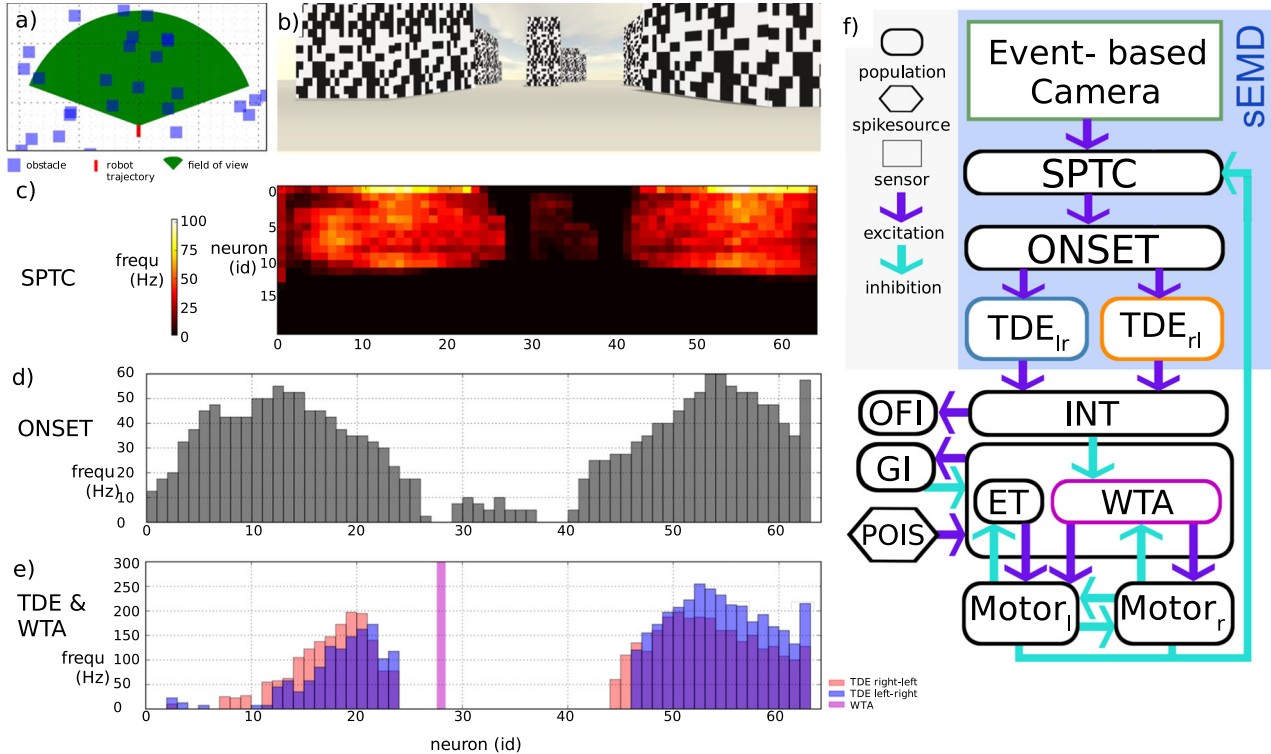

**Fig. 1 | Working principle of the obstacle avoidance network is demonstrated in an example run in a cluttered environment.** Robot is moving in a straight line with a velocity of 15 a.u. s⁻¹ (5 m s⁻¹). **a** Top view of robotic agent in the environment. **b** Robot camera view. The sky pattern is not visible to the robot. The ground is kept free of textures since it would cause events over the whole horizontal field of view. This problem can be avoided by removing the ground from the agent's field of view. **c** Spiking activity of the SPTC layer. It removes camera noise by bypassing spatio-temporally correlated events. **d** Spiking activity of the Onset (ONSET) layer. It reduces the 2D retinotopical map to a 1D horizontal map. Furthermore, it strengthens edge detection and reduces the number of spikes by self-inhibition. **e** Spiking activity of the TDE and WTA. TDE translate the time-to-travel between two adjacent pixels into a spiking rate. The single WTA spike indicates an obstacle-free direction. This spike activates the agent's next turning movement (saccade) by exciting one of the two motor populations in (**f**). The location of the winning neuron defines the direction and duration of the upcoming turn. **f** Obstacle avoidance network with spiking Elementary Motion Detectors (sEMDs) consisting of an event-based camera, spatiotemporal correlation (SPTC) population, ONSET population and two Time Difference Encoder (TDE) populations. Event-based camera (sensory input), SPTC population (noise filter and downsampling), ONSET population (reduces 2D retinotopical map to 1D and reduces the number of spikes by self-inhibition), TDE populations (time-to-travel translation to spike rate and inter-spike-interval (ISI)), Integrator (INT) population (spatial convolution from 64 to 16 neurons and slow dynamics due to long time constants), inverse Winner-Take-All (WTA) population (detects a minimum of OF, hence the gap), Escape Turn (ET) population (triggers turn when inverse WTA can not find direction), Motor (MOT) populations (control turn direction and duration), Optic Flow Integrator (OFI) population (modulates robot velocity), Poisson Spike Generators (POIS) (drive decision process with Poisson spike process) and Global Inhibition (GI) population (suppresses losing neurons in inverse WTA population and ET population).

tuning curve peaks at 1 Hz in a walking state, the sEMD's preferred direction velocity tuning curve is similar but peaks at 5 Hz. This suggests that the reported parameter set of the sEMD tunes it to slightly higher relative velocities. The model performs in a robust way for a wide range of illuminations (from 5 lux to 5000 lux) and relative contrasts (50% response reached approximately 35% relative contrast), as shown in Supplementary Fig. 2. A qualitative comparison with biological data supports our hypothesis that the sEMD approximates the elementary motion processing in the fly brain (see comparison between Fig. 2a and b). This processing is part of the input to the flight control and obstacle avoidance machinery. Hence, it can be used as an input for determining a collision-free path.

## Agent's behaviour

The robot's obstacle avoidance performance was evaluated in an experiment with the agent moving through environments with varying obstacle densities. Two more experiments were designed to further understand the mechanisms underlying the robot's movement performance. The agent's gap-crossing behaviour and tunnel-centring behaviour were investigated. These behaviours were analysed in insects in a plane. Accordingly, we limited our agent to a 2D motion.

## Agent's behaviour in corridors (real world)

One common experiment to characterise an agent's motion response to visual stimuli is to observe its centring behaviour in a tunnel equipped with high contrast patterns on the walls. The simple geometry of the environment enables the observer to directly relate the visual input with the agent's actions. In bees and flies, an increase in flight velocity proportional to the tunnel width has been observed[16,38,39]. In very narrow tunnels, insects show a pronounced centring behaviour, which decreases with increasing tunnel width. To prove the real-time capability and robustness of the SNN on neuromorphic hardware, we evaluated the system in a real-world scenario. Similar to the corridor experiment with bees and flies, the robotic platform described in Section "Robotic agent (real world)" was tested in a narrow (approximately 30 cm wide) and a wide corridor (approximately 50 cm wide; see Supplementary Fig. 11). The robot itself is approximately 20 cm wide. The robot centred well in nine out of ten runs in the wide corridors (see Fig. 3e). In the remaining run, the robot crashed into the wall at the very beginning since it was not able to find an obstacle-free heading direction (see Supplementary Fig. 12b). In another run, the robot did a 360° turn close to the end of the corridor. In the narrow corridor, the robot never crashed directly into the wall (see Fig. 3d) since the WTA always chooses an obstacle-

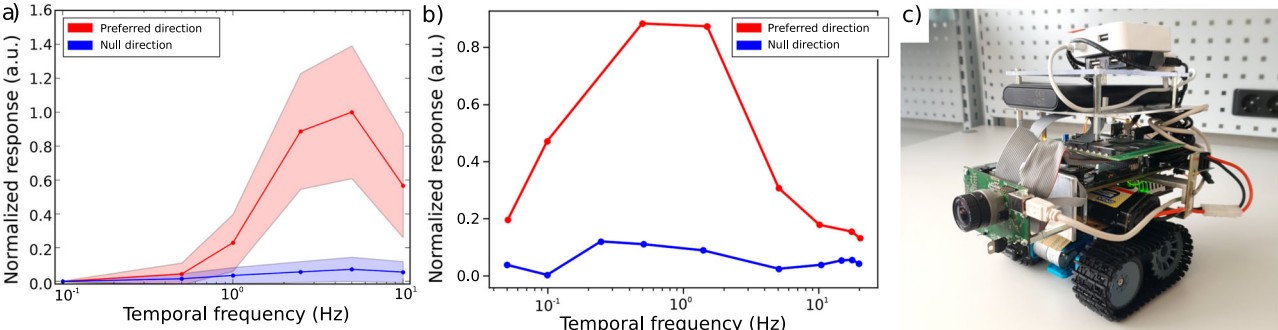

**Fig. 2 | sEMD response, *Drosophila* T4/T5 neuron response and robotic agent.** **a** Response of sEMD Preferred direction population and Null direction population in the Nest simulator to a square wave grating moving from top to bottom recorded with the PAER128 Event-based sensor. For sEMD response on the SpiNNaker neuromorphic hardware platform, see Supplementary Fig. 2. **b** Response of Preferred direction and Null direction T4/T5 neurons in *Drosophila* when presented with a square wave grating similar to (**a**). Data from Haag et al.[37]. **c** Robotic platform with PAER128 Event-based sensor, SpiNN-3 board and AERnode FPGA board for communication between the sensor, SpiNNaker board and motor controller. A small router is used to load code onto the SpiNNaker board, and a power bank and a battery pack power the hardware and the motors, respectively. The whole robotic system weighs 2.3 kg.

free movement direction (see Fig. 3g). In two out of these runs, the agent slightly touched the left wall. We observed an overall tendency toward the left of the corridor. A slight miss-alignment between the robot's field of view and its movement direction can explain this tendency. In the control experiment in which the obstacle avoidance population of the SNN did not receive any visual input, the robot turned directly to the left or right. It crashed into a wall in nine out of ten cases (see Fig. 3f). In the tenth case, the robot meandered through half of the tunnel before it collided with a wall. This control experiment shows that the visual input itself and not the intrinsic movement behaviour of the robot drives the centring behaviour.

## Agent's behaviour in corridors (simulation)

After successfully testing the corridor centring behaviour in the real world, we also evaluated the agent's performance in simulation. We performed experiments in three tunnels with different tunnel widths. Similar to the biological role model, the robot's velocity stands in a positive linear relationship with the tunnel width (see Supplementary Fig. 9). The measured velocity in a.u. per second is $1.25 \pm 0.13$, $1.62 \pm 0.64$, $2.00 \pm 0.83$ and $2.31 \pm 0.93$ for tunnel widths of 13.3, 16.6, 20, and 23.3 a.u., respectively. Furthermore, the robot always stays in the centre of the tunnel, especially in very narrow tunnels (see Fig. 3c). The deviation from the tunnel centre increases with the tunnel width (for the simulated robot, see Fig. 3a–c, physical robot see Fig. 3d–e), similar to observations in blowflies[38].

## Agent's behaviour in densely cluttered environments (simulation)

We evaluated the agent's obstacle avoidance performance in an arena with an obstacle density (obstacle density: percentage of the total area covered with objects.) between 0 and 38 % (0.05 objects per square a.u.) (see Supplementary Notes 3 and Supplementary Notes 4 for more details). The simulation stops either when the robot collides with an obstacle (collision: simulated robot's outline overlaps with the area occupied by object), when it leaves the arena, or when the simulation real-world-time of 6 h is over (approximately 1 minute of simulation time depending on the intensity of computation required). At low obstacle densities (<5%), many collision-free paths exist. The robot exhibits a random walk when the decision-making inverse WTA neuron population does not receive sufficient sensory drive. The inverse WTA receives spikes sampled from a Poisson process (see Figs. 1d and 4a). The resulting activity is dominated by this background activity, and the inverse WTA thus selects a winning neuron, i.e. a new heading direction, at random. This interplay of the Poisson background drive and feed-forward-driven OF results in a probabilistic decision process. The

decisions made by the network become less probabilistic with increasing obstacle density since the robot starts to follow the locally low object-density paths forming in the environment (see Fig. 4b). At obstacle densities higher than 20%, most of the gaps in the environment are smaller than the robot's minimum mean obstacle clearance (obstacle clearance: robot's distance to the centre of the closest object.) so that the agent stays close to its start location (see Supplementary Fig. 5 and Fig. 4c). A collision of the robot is generally caused by the robot's long reaction time in an environment with low mean obstacle clearance, equivalent to a high obstacle density (see Supplementary Fig. 5). Since the robot only senses visual stimuli in a 140° horizontal visual field, symmetrically centred around its direction of motion, there is a blind-spot behind the agent. After a strong turn, the simulated robot might be confronted with a previously not seen object and directly crash into it. Nevertheless, the agent shows a very robust obstacle avoidance behaviour in a large range of different environments with obstacle densities between 0 and 38%. The robot's mean success rate amounts to 97%.

While local OF is instrumental in finding gaps, global OF provides information about the clutteredness of the environment. Flies and bees decrease their flying speed when the clutteredness of the environment increases[38,39]. Our agent regulates its speed based on the global OF and consequently, it moves slower in denser regions of the environment (see Supplementary Fig. 7). To examine the effect of the velocity dependency, we ran a second experiment with the robot moving with constant velocity (see Fig. 4e and Supplementary Fig. 6). While with the velocity control only few collisions were encountered, for obstacle densities higher than 24% the number of collisions significantly increased when the velocity was kept constant.

## Agent's behaviour in gaps (simulation)

When presented with a choice between two gaps of different sizes, bees prefer to pass the larger gap[6,9]. This behaviour decreases the insect's collision probability significantly. Our agent chooses a larger gap by using a simple probabilistic integration mechanism, which is explained in the following. These findings imply that bees might also use a simple mixture of goal-directed movement and probabilistic exploration rather than complex decision-making to choose a gap[8].

In our setup, the simulated robot's upcoming movement direction is determined by an inverse WTA spike occurring in an obstacle-free direction, as shown in Fig. 1a–c. The exact functionality of probabilistic decisions in an inverse WTA is further explained in the Methods in Section "Obstacle avoidance network" paragraph 4. When confronted with a small and a large gap the probability of an inverse WTA spike appearing in the greater gap is higher. Hence, we assume

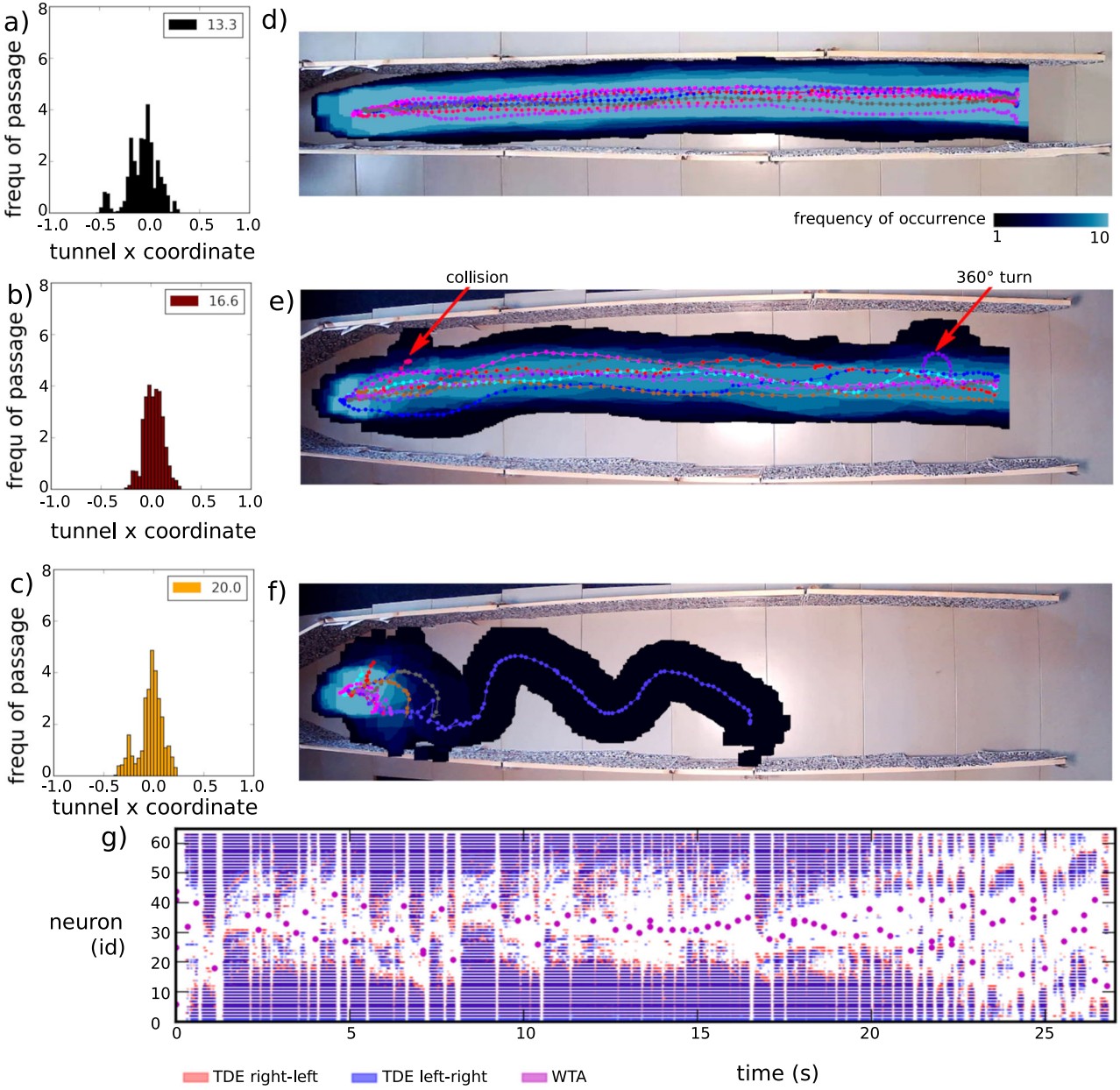

**Fig. 3 | Corridor centring experiments. a–c** Simulated corridor centring results normalised over the width of the corridor increasing from (**a** to **c**). The legend shows the corridor width in a.u. **d–g** Real-world corridor centring experiment results from the setup shown in Supplementary Fig. 11. **d** Robot's movement trajectories for ten runs through a narrow corridor moving from left to right. Dots and lines indicate the centre of mass of the robot. The blue area represents the whole area covered by the approximately 20 cm wide robot combined for all ten runs. The frequency of occurrence increases from dark blue to light blue. **e** Robot's trajectories for ten runs in a wide corridor. **f** Control experiment. Robot's trajectories in a wide corridor with no visual input. **g** Spiking activity in the robot's obstacle avoidance network during an example run in the narrow corridor. TDE spikes indicate the walls, and WTA spikes indicate the next turn of the robotic agent.

that the robot automatically follows pathways with a larger gap size. To evaluate this assumption we observed the robot's gap-crossing behaviour in an arena with two alternative gaps (see Fig. 4d). The robot can decide to cross any of the two gaps or stay in one-half of the arena. There is a competition between staying in the open space and crossing a gap. The larger the gap size is, the more likely the robot will cross a gap. We investigated the probability of crossing gaps by having two gaps, one with a fixed gap size (10 a.u.) and the other with a gap size between 3 a.u and 17 a.u. We calculated the gap entering probability by comparing the number of passes through both gaps. As expected the entering probability increases with gap size (see Fig. 4f). However, for smaller gap sizes, the probability of a spike pointing towards open space in the inverse WTA becomes significantly higher. Therefore, the

robot prefers to pass through gaps of larger size. Besides the gap width, the arena size changes the passing probability. In a smaller arena, the simulated robot stays closer to the gap entry, which increases the relative gap size sensed by the agent. Therefore, a larger part of the vehicle's visual field is occupied by the gap entry, which increases the probability of a spike occurring in the gap area. When the obstacle density exceeds 20%, most gaps fall below the gap entering threshold so that the robot can not leave the arena anymore (see Supplementary Fig. 5a and Fig. 4).

## Discussion
Flying animals master the art of avoiding collisions in a variety of environments. The abilities of volant insects, such as bees and flies, to

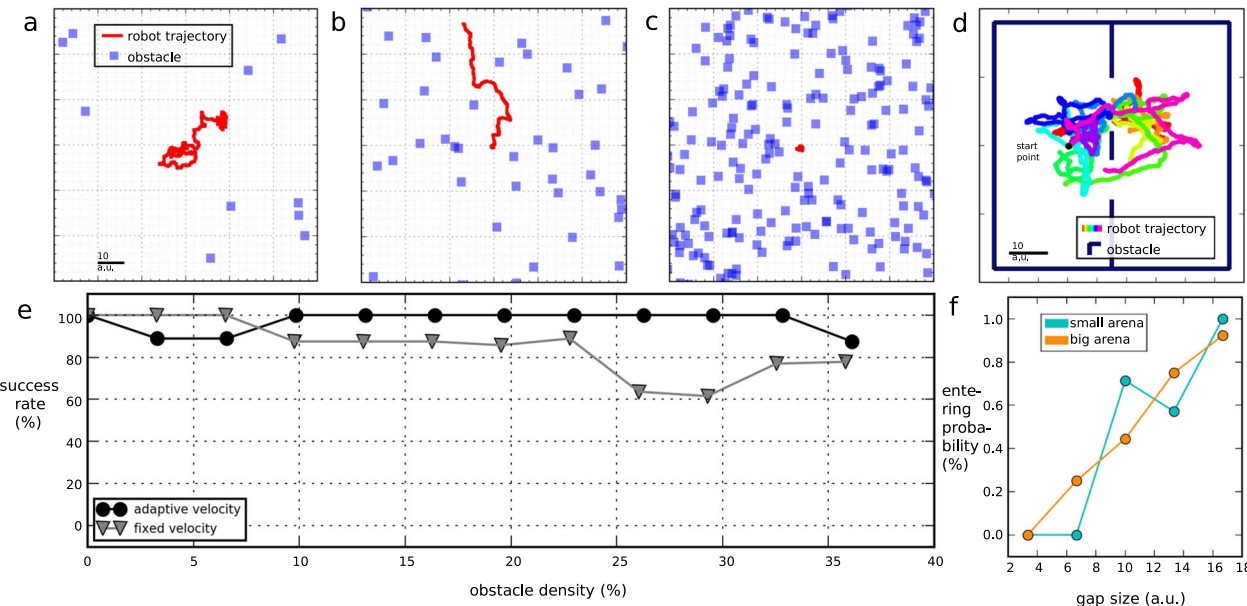

**Fig. 4 | Agent's behaviour in different simulated environments. a–c** Trajectories recorded in arenas with increasing obstacle densities. The first two seconds of the trajectory in (**a**) are shown in Supplementary Fig. 8 to illustrate the saccadic movement strategy of the robot. **d** Simulated robot's trajectory in the gap crossing experiment in a large arena. Colour represents time ($t_0$: light blue, $t_{end}$: magenta). **e** Simulated robot's performance in cluttered environments as shown in **a–c** with modulated velocity (black) and fixed velocity (grey). Agent's success rate, hence the number of runs without collisions. 79 runs were performed for the experiment with adaptive velocity, and 56 runs for the experiment with fixed velocity. 1 a.u. = 30 cm. **f** Gap crossing probability in dependency of the gap width for a large arena in (**d**) and a small arena.

manoeuvre in various environments have been studied systematically. Several mechanisms have been proposed to explain flight control and collision-free navigation in cluttered environments. Nevertheless, no mechanism has yet been found that can explain how insects solve navigational challenges, ranging from maintaining a course to meandering without collision in complex environments by manoeuvring through gaps. Inspired by the fly's neuronal machinery, we demonstrated a system-level analysis of a distributed, parallel and asynchronous neural algorithm mimicking several aspects of obstacle avoidance in insects. Our algorithm has the potential to be integrated into goal-directed routines. Our network comprised approximately 4k neurons and 300k synapses, which is well below the number of neurons in an insect brain. The agent guided by the algorithm robustly avoided collisions in various situations that are also solved by flying insects, from centring in a tunnel to crossing densely cluttered terrain by finding gaps. These behaviours were accomplished with a single set of parameters that were not optimised. We discuss our results in relation to the pertinent biological and engineering research.

Our agent showed many similarities to flying insects in its behaviour in spatially constrained environments. It meandered in cluttered terrains (Section "Agent's behaviour in densely cluttered environments (simulation)"), selected gaps (Section "Agent's behaviour in gaps (simulation)"), modulated its speed as a function of object proximity (Section "Agent's behaviour in corridors (simulation)") and centred in tunnels (Section "Agent's behaviour in corridors (simulation)") while using a saccadic control strategy (Section "Obstacle avoidance network")[9,16,38–40].

Navigating through cluttered terrain involves avoiding obstacles ahead selecting possible passages between obstacles while maintaining the intended overall course. The selection of passages can be reduced to a binary choice paradigm (e.g. left vs. right, up vs. down, etc.). In such experimental paradigms, flying bees tend to prefer the larger of the two gaps, but this choice also depends on individuality[9] and the aerodynamical forces constraining the generation of movements[11]. Like bees, our agent selected the larger of the two gaps (Section "Agent's behaviour in gaps (simulation)"). Gap crossing requires bees and our agent to detect free passages. Such passages should only be passed when the agent's size is not wider than the gap width. Our agent crossed gaps of 5 times its body width. In contrast, bees cross a single gap size as small as 0.5 times their wingspan but travel such gaps sidewise because their body length is shorter than their wingspan. They traverse gaps forward as small as 2.5 times their wingspan[8]. The discrepancy between our agent and the bees might be the consequence of different active vision strategies observed in bees and implemented in our agent. Our agent moved according to an active gaze strategy of saccadic turns interspersed by translation and relied solely on OF. However, bees assess the possibility of crossing a gap by performing sideward scanning manoeuvres before they start crossing it. The more difficult the gap is to cross, the longer are such manoeuvres[8]. In addition to optic-flow cues, flying insects use the brightness within the gap relative to the brightness surrounding the gap[6]. The combination of brightness and optic-flow cues may help the bees steer through small gaps.

Our agent moved through cluttered environments with an obstacle density between 0 and 38%, with 97% of collision-free travels. We examined the simulated robot's performance to understand the essential behavioural components which led to such a low collision rate. The most significant ingredient was the implementation of the dependency of locomotion velocity on the strength of the global OF. This insect-inspired regulation of velocity from optic flow improved the obstacle avoidance performance of the agent from a mean success rate of 85% to 97% (see Fig. 4e). We propose that this kind of velocity controller could be regulated in insects by a simple feedback control loop. This loop changes the agent's velocity inversely proportional to the global OF integrated by a subset of neurons (see Section "Obstacle avoidance network").

To traverse both cluttered and bare terrains, flying insects require maintaining an intended course. Course control in flying insects has been studied by letting insects travel along a corridor. The insects tend to centre in the middle of the corridor width[16]. This behaviour is suggested to be based on balancing the OF on both eyes[17]. Our agent also maintains a centred course in a corridor without implementing a

mechanism of OF balancing. It seems that the agent's centring behaviour might be due to moving in the direction of the largest time to contact in the visual field (i.e. the region of the lowest apparent motion). Further behavioural experiments with volant insects are required to disambiguate between optic-flow balancing or following areas of low apparent motion. By manipulating the OF perceived by the insects either in virtual reality (e.g.[41,42]) or mechanised environments, one could create a conflict between the two hypotheses and clarify the strategies used by volant insects.

Our model shares several similarities with the neural correlates of visually-guided behaviour in insects, including motion-sensitive neurons[27], an integration of direction[43], an efference copy mechanism at the level of integrating neurons (saccadic inhibition from the motor neurons to the inverse WTA and visual input neurons)[44], and neurons controlling the saccade amplitude[40]. Our agent adopted an active gaze strategy due to a saccadic suppression mechanism. When the inverse WTA layer in the model did not "find" a collision-free path (i.e. a solution to the gap-finding task), an alternative response (here a U-turn) was triggered thanks to global inhibitory neurons and excitatory-inhibitory networks (GI-WTA-ET, for more details, see Section "Obstacle avoidance network"). The neuronal correlate of such a switch has not yet been described, to our knowledge, for flying insects. Our model, thus, serves as a working hypothesis for such a neuronal correlation.

The hardware used in this work, i.e. event-based sensors and spiking neural networks, provide a powerful computational framework to explore the principles of insect-inspired obstacle avoidance on a real robotic platform. Event-based sensors inherit precise timing information in their sparse output, which enables real-time OF computation using a simple elementary motion detector model. The output of event-based sensors consists of short digital electric pulses. This event data stream does not directly mimic the information processing in the insect neural system, which is mainly based on graded potentials continuous analogue signals. An expensive and error-prone conversion to analogue signals, which are closer to the information processing in insects, does not appear useful[45,46]. The whole obstacle avoidance network uses action potentials, successfully replicating the response of motion-sensitive neurons and overall insect behaviour on a high level of abstraction. This approach has the appealing advantage of naturally leading to the extraction of computational principles rather than putting emphasis on the precise emulation of single-cell functionality. Therefore, our implementation, based on the asynchronous processing of OF information, can be regarded as bio-inspired and used to study processes downstream of elementary motion detection.

Navigating insects are not solely guided by obstacle avoidance. Instead, they have a goal, such as finding a mate, food resources, or returning to their home. In contrast, our agent did not have such a goal, i.e. a specific point in space it was required to reach. Rather, our agent was attracted by regions in its visual field with low apparent motion. The absence of a goal may explain some of the differences observed at the behavioural level between our agents and insects. How might the implementation of goal-directed behaviour change our agent's performance while crossing corridors, gaps, or cluttered terrain?

When the goal of flying insects is not located at the corridor centre but on one side of it, they move along the wall[47]. In this situation, the OF on both sides of the visual field is not balanced; rather, the OF provided by the distant wall does not guide the insect's movement in the tunnel while approaching the goal but only that of the close wall, i.e. the wall eliciting the faster OF. While our current agent centres in a tunnel, extending our model with a goal direction would probably lead to a similar wall-following behaviour. Such a behaviour will be explored in future work.

We also observed that our agent crosses only gaps of at least five times the agent's body size, in contrast to the 2.5 times observed in bees. A gap smaller than five times its body size was probably not yielding sufficiently low regions of apparent motion in the agent's visual field. When confronted with a smaller gap, the agent moved in another direction rather than into the gap region. The reported behaviour of bees while crossing gaps focused on the foragers. These individuals are motivated to return to their nest. Therefore, the gap crossing behaviour in bees is likely to be a combination of following a goal direction (returning to their nest) with avoiding a collision with the gap. This might be accomplished by an algorithm proposed by Hoinville et al. based on Bayesian cue integration. This framework suggests that when two directions are aligned, the precision of the integrated direction is greater than the single direction (similarly to optimal cue integration). Thus, the integration yields a more precise input to the motor control, perhaps permitting the bees to cross small gaps.

Finally, agents driven by an obstacle avoidance routine display search-like behaviour in cluttered terrain. They meander and circle around the objects and do not move in an overall direction. By combining such a routine with an overall direction, agents may display a route-following-like behaviour through the clutter[18]. Our agent, guided only by an obstacle avoidance routine, also displayed a search-like behaviour in clutter. A first open-loop approach combining our obstacle avoidance network with a goal direction provided already promising results[48]. Thus, our agent may display behaviour closer to volant insects by providing it with a goal to move towards.

Neuromorphic systems such as the robot used in this project make use of the main principles of brain computing identified until today. On the one hand, neuromorphic systems are a promising candidate for robotics and computing on the extreme edge. These systems have been shown to work power-efficient and in computational real time. On the other hand, neuromorphic systems can be used to understand how the brain works on a certain level of abstraction. In this article, we aim to understand how obstacle avoidance could be performed in flies and bees based on a gap-finding strategy rather than actively avoiding obstacles. To date, most other insect-inspired obstacle avoidance approaches rely on traditional cameras[18,49–53]. Some approaches investigate algorithms for obstacle avoidance in open-loop using event-based cameras[32,54–61] (for a more detailed comparison of mentioned approaches, refer to Milde et al.[32]). Closed-loop obstacle avoidance behaviour has been demonstrated previously using fully conventional sensory-processing (frame-based sensor and CPUs/GPUs) approaches[52,53] (for extensive review please refer to Serres et al. and Fu et al.[17,62]). Mixed-system approaches using event-based cameras and conventional processing can also be found[59,63]. However, only very few fully neuromorphic closed-loop systems exist[64,65]. To our knowledge, the system presented in this article is the very first extensive study on obstacle avoidance, which implements a whole system from sensing until actuation purely in events and spikes. The system incorporates almost all principles of spike coding in just a single neural network. The obstacle avoidance network receives spike timing information from the sensory input. It encodes the spike timing into a rate code projecting onto a one-dimensional retinotopic map. Spatial coding throughout almost all neural network populations encodes the position of the visual input as well as the intended movement direction. Further mechanisms such as saccadic inhibition, proprioceptive feedback and behavioural switches further increase the richness of the network. Thus, this extensive study on obstacle avoidance is the first insect-inspired closed-loop approach that demonstrates the full potential of fully spike-based sensorimotor systems. With a success rate between 90% and 100%, our system shows similar or even better performances than state-of-the-art obstacle avoidance approaches (see Table 1). The computational pipeline runs

**Table 1 | Comparison of different insect-inspired obstacle avoidance approaches**

| Author | Sensor | Capability | Scenario | Environment | Vehicle | Success rate | Algorithm |
|---|---|---|---|---|---|---|---|
| Our work | Camera | OA | Clutter | Sim | Wheeled robot | 90–100 (36–0% OD) | OF SNN, adap. velocity |
| Our work | Camera | OA | Corridor | Real | Wheeled robot | 95 | OF in SNN |
| Lu[85] | Depth | OA | Clutter | Sim | MAV | 82–95 (velocity 2.25% OD) | OFM |
| Lu[85] | Depth | OA | Clutter | Sim | MAV | 81 (2.25% OD) | DDPG |
| Lu[85] | Depth | OA | Clutter | Sim | MAV | 87 (2.25% OD) | NAF |
| Cho (2019) | Camera | OA | Junction, Corridor, Ramp | Sim/real | MAV | – | 3D OF balancing |
| Nous (2016) | Camera | OA & goal | Clutter and goal | Sim/real | Drone | 0–100 (20–1% OD) | Force field |
| Nous (2016) | Camera | OA & goal | Clutter and goal | Sim/real | Drone | 20–100 (20–1% OD) | Potential field |
| Nous (2016) | Camera | OA & goal | Clutter and goal | Sim/real | Drone | 20–100 (20–1% OD) | Rule based |
| Bertrand (2015) | Camera | OA & goal | Clutter and goal | Sim | Circle | – | Nearness vector from OF |
| Fu (2016) | Camera | OA | Clutter | Sim/real | 2-wheel-robot | 85–95 (threshold tuning) | LGMD SNN |
| Zingg (2010) | uEye camera | CC | Corridor | Sim | MAV | – | OF balancing |
| Hyslop (2010) | Camera | OA | Urban environment | Sim | MAV | – | WFI |
| Sanket (2018) | Camera | GD | Gap | Real | Quadrotor | 85 | $TS^2P$ |

*OA* obstacle avoidance, *CC* corridor centring, *OD* obstacle density, *OF* optic flow, *GD* gap detection, *SNN* spiking neural network, *OFM* only forward manoeuvre, *DDPG* deep deterministic policy gradients, *NAF* normalised advantage functions, $TS^2P$ temporally stacked spatial parallax, *WFI* wide field integration, *LGMD* lobula giant movement detector.

on 1–4 Watts (DVS128 PAER: 23 mW[66], SpiN-3: 1–3.8W[67], AerNode: 30 mW for Neuromorphic Auditory Sensor[68], implementation dependent), which is four times more energy efficient than a comparable approach on a Nvidia TX1 GPU using 6–16 Watts[69]. At the same time, it is the first algorithm which can perform gap crossing, obstacle avoidance in cluttered environments and corridor centring just using one single set of parameters. This algorithm is a promising implementation that should be further explored in natural environments. Natural environments lead to new challenging scenarios, such as low-contrast walls, which cannot be detected by event-based cameras. The optic flow of such areas can be predicted by interpolating the shape and area of the object from the events at the edges. Another option would be to add a different type of sensor, such as radar, which is able to detect structureless objects. By extending our work in these directions we can make it even more robust for real-world applications. Thus, this neuromorphic system brings us one step closer towards exploring the full potential of Neuromorphic hardware for robotics.

## Methods

Most experiments in this article were conducted in simulation using the nest spiking neural network (SNN) simulator[36] and the Neurorobotics Platform environment[34] (parameters see Supplementary Table 1). A corridor-centring experiment was conducted in a real-world corridor using a robotic platform equipped with an event-based camera (the dynamic vision sensor[22]) as visual input and a SpiNN-3[70] board for SNN simulation in operational real-time. Sensory data for the sEMD characterisation (see Section "Spiking elementary motion detector") were recorded with an event-based camera in a real-world environment. The hardware, software, SNN models and methodologies used in this article are explained in the following.

### Spiking neural networks and event-based sensors

The insect brain is a great role model for novel emerging technologies. The massively parallel recurrent 3D network with localised memory is able to perform complex tasks such as navigation in operational real-time while requiring resources only in the range of milliwatts. Neuromorphic hardware is one emerging technology which aims at implementing principles of brain computation in CMOS and novel devices. Neuromorphic systems are hardware implementations of massively

parallel networks of neurons and synapses[71]. Today, a great variety of neuromorphic hardware can be found, ranging from asynchronous subthreshold mixed analogue/digital CMOS implementations[72,73] up to digital, clocked, spiking neural network simulators[74]. While different approaches benefit from different aspects of brain-inspired computing, such as asynchronous computation[72,73], sparsity, local memory[75,76] and low power consumption all of them aim to get one step closer towards brain-inspired computing. In combination with event-based sensors, neuromorphic hardware paves the way for a new generation of brain-inspired systems. Event-based sensors sample changes in the environment and only convey novel information to the sensory output. This kind of sampling reduces the amount of data and, therefore, the required computation and related power consumption while increasing the temporal resolution. At the same time, event-based sensors are ideally suited as input to spiking neural networks. Both event-based sensors, as well as spiking neural networks carry information in the form of short digital pulses. In this article, we used the DVS PAER128, an event-based vision sensor, as input to the spiking neural network. We collected real-world data using the event-based camera[22] to characterise the sEMD response. The sensor comprises $128 \times 128$ independently operating pixels which respond to relative changes in log intensity, i.e. temporal contrast. When the change in light intensity exceeds an adaptive threshold the corresponding pixel produces an event. The address of the pixel and polarity are communicated through an address event representation bus[77]. Light increments lead to ON-events, whereas light decrements lead to OFF-events (polarity). The sensor reaches a dynamic range of more than 120 dB and is highly invariant to the absolute level of illumination due to the logarithmic nature of the switched-capacitor differencing circuit[22,66].

### Spiking elementary motion detector

In 2018, a new insect-inspired building block for motion vision in the framework of SNNs was proposed. This building block is designed to operate on the output event stream of event-based cameras and is called the spiking elementary motion detector (sEMD)[32]. The sEMD is inspired by the computation of apparent motion, i.e. optic flow (OF), in flying insects. In contrast to its gradient-based role model, the sEMD is spike-based. It translates the time-to-travel of a spatiotemporally correlated pair of events into a direction-dependent output burst of

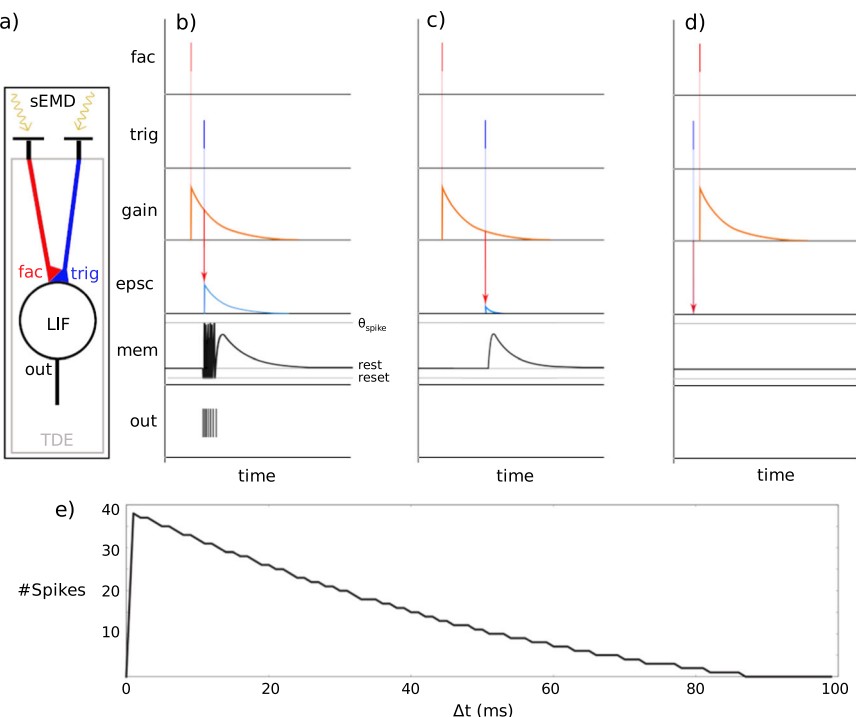

**Fig. 5 | Spiking elementary motion detector model adapted from[84]. a** sEMD model consisting of visual input and TDE unit. Two adjacent retina inputs are connected to the facilitatory synapse (fac) and the trigger synapse (trig), respectively. The facilitatory synapse controls the gain of the trigger synapse's postsynaptic current (epsc), which integrates into the LIF neuron's membrane potential, which produces output spikes (out). **b** Model behaviour for small positive Δt. **c** Behaviour for large positive Δt. **d** Behaviour for negative Δt. **e** Number of output spikes over Δt.

spikes. While the sEMD provides OF estimates with higher precision when the entire burst is considered (rate code), the interspike interval distribution (temporal code) within the burst provides low-latency estimates. The sEMD consists of two building blocks: a retina to extract visual information from the environment and the TDE, which translates the temporal difference into output spikes (see Fig. 5a). When the TDE receives an input spike at its facilitatory pathway, an exponentially decaying gain variable is generated. The magnitude of the synaptic gain variable during the arrival of a spike at the trigger synapse defines the amplitude of the generated excitatory post-synaptic current, thus implementing direction sensitivity. This current is integrated into the sEMD's membrane potential, which generates a short burst of output spikes. Therefore, the number of output spikes inversely encodes the stimulus' time-to-travel (see Fig. 5e) between two adjacent input pixels. The motion detector model has been implemented and evaluated in various software applications (Brian2, Nengo, Nest), on neuromorphic digital hardware (SpiNNaker, Loihi), on FPGA and also as an analogue CMOS circuit[32,48,78,79].

### Obstacle avoidance network
The obstacle avoidance network infers a collision-free direction from its sEMD outputs and translates this spatial information into a steering command towards open space (See Fig. 1d and Supplementary Fig. 3 for a more detailed version, network parameters and connections see Supplementary Tables 4–7). The first layer, the event-based camera, generates an event when a relative change in log illumination, i.e. temporal contrast, is perceived by a pixel (see Fig. 1b). A macropixel consists of 2 × 2 event-based camera pixels. Each macropixel projects onto a single current-based exponential leaky integrate and fire (LIF) neuron (hereafter referred to as LIF for the sake of clarity) in the spatiotemporal correlation (SPTC) layer (in Nest the neuron model used throughout this study is called iaf_psc_exp). Each SPTC neuron emits a spike only when more than 50% of its receptive field elicits an event within a rolling window of 20 ms. This mechanism is implemented by setting the spiking threshold high enough so that at least two consecutive spikes are needed for the membrane potential to reach the threshold. The membrane potential quickly decreases after every spike due to a leak. Hence, the spikes have to happen in a short time period to reach the threshold. Therefore, the SPTC population removes uncorrelated events, which can be interpreted as noise. Additionally, it decreases the network resolution from 128 times 40 pixels to 64 times 20 neurons.

The next layer extracts OF information from the filtered visual stimulus. OF is described as the apparent motion of objects over the visual field of a moving agent. While translational OF is inversely proportional to the relative distance of objects, rotational OF does not contain any depth information. Hence, inspired by findings in insects, we have divided the agent's movement behaviour into two phases, intersaccades during which the agent moves straight forward and extracts depth information, and saccades during which the agent turns towards the new movement direction (see Supplementary Fig. 8). The layer in our network which extracts OF consists of two TDE populations sensitive to the two horizontal cardinal directions respectively. Each TDE receives facilitatory input from its adjacent SPTC neuron and trigger input from its corresponding SPTC neuron. The facilitatory input might arise either from the left (left–right population) or from the right (right–left population). The sEMD output encodes the OF as a number of spikes in a two-dimensional retinotopical map (see Fig. 1c).

Since the agent moves on the ground it only estimates the amount of horizontal OF. Hence, the subsequent Integrator (INT) population integrates the spikes of each sEMD column in a single LIF neuron. This reduction to a one-dimensional map reduces the number of neurons needed in the subsequent layers by a factor of 40.

The subsequent population, an inverse soft winner-take-all (WTA), determines the agent's movement direction towards a minimum of OF in the one-dimensional retinotopical map (see Fig. 1c, purple stripe).

Since OF encodes the relative distance of objects during a translational movement, this direction represents an object-free pathway. The inverted WTA receives inhibitory input from the INT population, not excitatory input, as done in a classical WTA network. Hence the term inverted. The Poisson spike generators (POIS) inject Poisson-distributed background spikes, ensuring that one neuron within the inverse WTA wins at any moment in time, even in the absence of OF. In the absence of INT input, the inverse WTA neuron with the strongest POIS input wins and suppresses the activity of all others through the global inhibition (GI) neuron, leading to random exploration. Local lateral excitatory connections in the inverse WTA population strengthen the winning neuron (for the sake of clarity, recurrent excitation is not shown in Supplementary Fig. 3). Due to the consistently changing nature of the POIS spike trains the winner changes frequently, and the agent executes a random walk (see Fig. 4a). When the agent approaches an object the position and relative distance of the obstacle is indicated by a number of spikes in the INT population. These spikes strongly inhibit the inverse WTA at the corresponding position and its closest neighbours so that this direction cannot win. Therefore, the active neurons in the inverse WTA always represent an obstacle-free direction. In case no object-free direction has been found for approximately 220 milliseconds since the start of an intersaccade, the straight movement phase during which the agent collects distance information, the escape turn (ET) neuron emits a spike. This neuron is only weakly excited by the POIS population and connected to the GI neuron, similar to the inverse WTA population. The ET neuron only wins when it has not been inhibited for a long time; hence, the inverse WTA was not able to generate a spike due to strong overall inhibition.

The final layer, called the Motor (MOT) population, translates the activity of the inverse WTA population and the ET neuron into a turn direction and duration using pulse-width modulation to control the motors. During the inactivity of the MOT population, the last pulse-width modulation signal drives the motors. The left turn MOT population is activated by inverse WTA neurons on the left side, and the right-turn population by inverse WTA neurons on the right side. Since the turning velocity is always constant, the angle of rotation is defined by the duration of the turn. This duration of the excitatory wave in the MOT population relates proportionally to the distance of the winning inverse WTA neuron from the centre of the horizontal visual field. The duration saturates for neuron distances higher than nine neurons. Since a left turn and a right turn are mutually exclusive events, strong inhibition between the two MOT populations ensures to disambiguate of the MOT layer outputs. In case the ET neuron emits a spike, the excitatory wave passes through most neurons of the left MOT population. Hence, the turning duration is slightly higher than for any turn induced by the inverse WTA population. The agent turns completely away from the faced scene since no collision free path was found in that direction.

During the execution of a turn, the gap finding network receives mainly rotational OF. This type of apparent motion does not contain any depth information, and therefore, no new movement direction should be chosen during or shortly after a turn. Therefore, the MOT layer strongly inhibits the inverse WTA and SPTC populations as well as the ET neuron. After a turn has finished and none of the MOT populations is spiking anymore, the agent moves purely translatory. The movement speed during this phase $v_{ints}$ is defined in Eq. (1) where $\bar{f}_{OFI}$ is the mean firing rate of the optic flow integrator (OFI) population. This guarantees that the velocity is adapted to the level of clutteredness, ensuring slow movement in complex environments. During this movement phase, called intersaccade, the agent integrates translational OF information in its INT population. The inverse WTA population slowly depolarises from its strongly inhibited state and releases a spike, indicating the new movement direction. This spike triggers the next saccadic turn of the robot while the identifier of the winning

neuron defines the direction and duration of the movement.

$$v_{ints}(ms^{-1}) = 1 - \bar{f}_{OFI} \times 0.001 \tag{1}$$

### sEMD characterisation (real-world recordings and SpiNNaker/Nest)
For the sEMD characterisation, we stimulated an event-based camera with a 79° lens (see Section "Event-based cameras in Gazebo (simulation)") using square-wave gratings with a wavelength of 20° and various constant velocities from 0.1 to 10 Hz. These recordings were performed in a controlled environment containing an event-based camera, an LED light ring and a moving screen which projects exchangeable stimuli (see Supplementary Fig. 1). The contrast refers to absolute grey-scale values printed on white paper to form the screen. However, given the printed contrast, we calculated the Michelson contrast as follows:

$$\frac{I_{max} - I_{min}}{I_{max} + I_{min}} = \frac{I_{max} - I_{min}(1 - C_{printed})}{I_{max} + I_{max}(1 - C_{printed})} = \frac{C_{printed}}{2 - C_{printed}} \tag{2}$$

To show the model's robustness to a wide range of environments, we varied the following three parameters in the recordings: The illumination, the grating velocity and the grating's contrast (see Supplementary Table 1). The event-based camera was biased for slow velocities. The sEMD model (see Supplementary Fig. 3, first three populations) was simulated on SpiNNaker and in Nest with the neuron parameters defined in Supplementary Table 3. The mean population activity of the preferred direction and null direction population were calculated (see Fig. 2a). For the closest comparability to the biologically imposed environment parameters, we chose to compare and discuss the sEMD's velocity tuning curve for a grating contrast of 100% and an illumination of 5000 lux.

### Robotic agent (real world)
We developed a physical robotic agent to validate the real-world applicability of our model (see Fig. 2b). The robot receives visual input from a Dynamic Vision Sensor with a horizontal viewing angle of 110°. The event-based camera sends its events to a SpiNN-3 board, which contains a simplified version of the obstacle avoidance network described in Section "Obstacle avoidance network". The network does not require any OFI neurons since the agent moves with a constant velocity of around 0.1 m s⁻¹. No ET population is included. The motor control is regulated by an FPGA-based AERnode board. The board receives input from one SpiNNaker output neuron population. It translates the spiking input into pulse-width-modulation signals to control the motors. The pulse width of the signal depends on the identifier of the output neuron spiking on the SpiNNaker board. Two pulse-width-modulation signals are used to move the robot in a differential steering mode.

### Event-based cameras in Gazebo (simulation)
Kaiser et al.[80] developed a Neurorobotics Platform implementation of an event-based camera based on the world simulator Gazebo. This model samples the environment with a fixed update rate and produces an event when the brightness change between the old and new frames exceeds a threshold. We used this camera model in our closed-loop simulations as visual input to the obstacle avoidance network. Even though Gazebo produces an event stream from regularly sampled synchronous frame difference, our sEMD characterisation and open-loop experiments (see Section "sEMD characterisation (real-world recordings and SpiNNaker/Nest)") confirmed the working principle of the motion detector model with real-world event-based camera data. The real-world fully-neuromorphic applicability in the closed-loop of most parts of the simulated agent, including the apparent motion

computation by the sEMD and the saccadic suppression, was also demonstrated previously[81]. We set the resolution of the Gazebo event-based camera model to 128 times 40 pixels. The reduction of the vertical resolution from 128 to 40 pixels was done to speed up the simulation time and to make the model fit onto a SpiNN-3 board[70]. To further accelerate the simulation, we limited the number of events per update cycle to 1000 and set the refresh rate to 200 Hz. Therefore, the sEMD can only detect time differences with a resolution of 5 ms. We decided for a large horizontal visual angle of 140° so that the robot does not crash into unforeseen objects after a strong turn. At the same time, the uniform distribution of 128 pixels over a 140° horizontal visual field leads to an inter-pixel angle of approximately 1.1°. This visual acuity lies in a biologically plausible range of inter-ommatidial angles measured in Diptera and Hymenoptera, which varies between 0.4° and 5.8° [82].

### Robotic agent in Gazebo (simulation)

We designed a four-wheeled robot model for the Gazebo robotics simulator. The robot's dimensions are $20 \times 20 \times 10$ cm, and it is equipped with an event-based camera (see Section "Event-based cameras in Gazebo (simulation)") and the husky differential motor controller plugin. The brain interface and body integrator (BIBI)[34] connects the robot with the obstacle avoidance network implemented in NEST (see Section "Obstacle avoidance network"). During the inactivity of the MOT populations, the robot drives purely translatory with a maximum speed of 2.5 a.u s$^{-1}$. The movement velocity changes inversely proportional to the environment's obstacle density, as explained in Section "Agent's behaviour in densely cluttered environments (simulation)". When one of the two MOT populations spikes, the robot fixes its forward velocity to 0.38 a.u s$^{-1}$ and turns either to the left or to the right with an angular velocity of 4° s$^{-1}$.

### Neurorobotics platform (simulation)

To perform our behavioural experiments we decided to simulate the entire system, from visual input to actions, using the Neurorobotics Platform. This platform combines simulated SNNs with physically realistic robot models in a simulated 3D environment[34]. The platform consists of three main parts: the world simulator Gazebo, the SNN simulator Nest and the Transfer Function Manager BIBI. The BIBI middleware consists of a set of transfer functions which enables the communication between Gazebo and NEST via robot operating system (ROS)[83] and PyNN adaptors. The closed loop engine (CLE) synchronises the two simulators, Gazebo and Nest, and controls the data exchange through transfer functions. The simulation front-end virtual coach is useful to control the whole simulation procedure through a single Python script. Furthermore, the State Machines Manager of the SMACH framework can be used to manipulate the robot or world environment during the experiment.

Five different environments were designed in the neurorobotics platform to evaluate the agent's performance: a cluttered environment with randomly distributed obstacles sizing $1 \times 1$ m, an environment with two arenas connected by two gaps of variable size, a tunnel with varying width, an empty box environment and a narrowing tunnel. No obstacles were placed in a radius of 2 m around the agent's start point so that the system could reach a stable state of activity before being confronted with the first object. At obstacle densities higher than 35%, the agent stays at its start point since no obstacle-free direction can be detected anymore. Therefore, we limited the tested obstacle density range to between 0% and 38%. All obstacles placed in the environment, including walls, were covered with black-and-white randomly structured gratings.

A state machine within the neurorobotics platform environment automatises the experiments. The state machine consists of eight states, as shown in Supplementary Fig. 4.

Additionally, a virtual coach script starts and stops the single runs in a for-loop. CSV files containing the spiking data of the network, the robot position and angular alignment as well as the placement of the objects in the arena, were saved for all experiments. 79 data points were collected for the obstacle avoidance experiment in a cluttered environment with adaptive velocity, and 56 data points were collected for the experiment with fixed velocity (see Fig. 4e and Supplementary Figs. 5–7). The tunnel centring experiment, gap entering experiment and all other simulation experiments in the supplementary material were repeated 10 times for each individual configuration (see Supplementary Table 2). The experiment in cluttered environments lasts 6 h, while all other experiments last 2 h (real-world time).

Obstacle densities were calculated by plotting the cluttered environment and counting the number of pixels occupied by the objects. The occurrence of collisions was also measured visually by plotting the cluttered environment with the robot's trajectory while considering the agent size and angular alignment. Since the *can_collide* feature of the objects in the cluttered environment was turned off the agent moves through the obstacles when colliding. Therefore, an overlap of obstacle and robot can be interpreted as a collision. The obstacle avoidance run was marked as failed when such an overlap occurred, and the first time of overlap was noted as collision time. Since there is no physical collision the robot's size can be varied during the analysis to evaluate the effect of agent size on the performance. To enhance the comparability of the robotic system to the biological role model, flying insects, we normalised all distance measures by dividing them by the chosen robot's size of $30 \times 30$ cm. The normalised distance measures were complemented with an arbitrary unit (a.u.).

## Data availability

The data generated during this study are available at *dataverse.nl/dataset.xhtml?persistentId=*https://doi.org/10.34894/QTOJJP.

## Code availability

The code generated during this study is available at *github.com/thorschoepe/collisionavoidance_NRP* and https://github.com/thorschoepe/collision_avoidance_SpiNNaker.

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

## Acknowledgements

The authors would like to thank Daniel Gutierrez-Galan and Florian Hofmann for their technical support. The authors would also like to acknowledge the SpiNNaker Manchester team for their help with the sEMD implementation and the robot setup. Furthermore, the authors acknowledge the Neurorobotics Platform team for their technical support. The authors would like to acknowledge the financial support of the CogniGron research centre and the Ubbo Emmius Funds (University of Groningen).

## Author contributions

T.S., E.C. and E.J. conceived and designed the experiments. M.B.M., T.S. and E.J. designed and optimised the tested algorithm. T.S. and E.J.

carried out and analysed the experiments. M.B.M. and E.C. developed the original sEMD model. O.J.N.B., M.B.M, T.S., E.J., E.C. and M.E. wrote the paper.

## Funding

## Competing interests

The authors declare no competing interests.
