## [Peer Review File · Nature Communications]

Finding the Gap: Neuromorphic Motion-Vision in Dense EnvironmentsREVIEWER COMMENTS

Reviewer #1 (Remarks to the Author):

The authors propose a fully neuromorphic pipeline for optic-flow-based obstacle avoidance on a robot, with spikes propagating from an event camera through a spiking neural network and finally to motor actuators. The obstacle avoidance scheme is quite advanced, including slowing down when the environment gets cluttered, and performing navigation in saccades – similar to how flies navigate. The robot’s “artificial brain” contains many elements that are non-standard in AI (besides the dynamics of spiking neurons, also inverse winner-take-all structures, etc.). The brain is completely handmade and tuned to exhibit desirable properties, such as passing through certain gap widths or turning away from them. These behaviors emerge from the neural parameters and dynamics. This is very interesting when compared to, e.g., common deep reinforcement learning practices.

In my opinion this work is very impressive and definitely forms the type of significant advance that Nature Communications intends to publish. Moreover, it is of relevance not only to robotics and engineering fields, but also to biology, as specific mechanisms are proposed that can potentially explain biological phenomena.

I do have a number of remarks that may be used to improve the manuscript.

Main remarks:

1. The advantages of neuromorphic pipelines would be energy efficiency and fast, asynchronous updates.
 - a. Can the energy expenditure of the neuromorphic pipeline be measured? How does it compare to, e.g., an Intel Realsense + NVidia Xavier? If measuring is not possible, can you estimate the energy costs? And what was the mass of all required components?
 - b. The vision (optic flow) seems a bit “slow”, tuned to 5Hz. But this may be an impression that is not true, since later (line 388) it is mentioned that the wave grating covers 20 degrees. Is 5 Hz then equal to 100 deg / second? In general, the authors use temporal frequency as a measure for optic flow speed, which is confusing in my eyes, because optic flow is an angular change over time – not a frequency. The authors would do well to clarify this matter.
2. The slowing down in cluttered environments is very interesting. Fig. 4e shows the result that goes with the finding that this helps to improve obstacle avoidance at high densities. However, this graph where a fixed velocity is better, between 10 and 20%. Moreover, whether one is better than the other in that graph is hard to evaluate, since there are no whiskers to indicate percentiles in the data. On how many runs are these results based? Is the reason for this that there are only three runs per condition? If so, would it be possible to do more runs?
3. Visualization of the direction selection: Fig. 1 shows the winning direction with a purple bar. However, it would be interesting to see the INT layer as well, which (I guess) shows a more continuous picture of

good and bad directions. Some more examples in the appendix would be appreciated. Fig 1 is very important, but in general could be improved: We see events at textureless locations (apparently these have grated patterns, but they are not shown). We do not see events at locations in the sky that have some cloud texture. The ground is textureless... I propose that the authors show texture on the walls. Moreover, the authors should comment on whether ground texture would interfere with their current scheme. I think it would, and although no showstopper (it can easily be ignored on a driving robot just in terms of image coordinates), it should be mentioned. Fig. 1c in my eyes almost shows random flow, whereas I would expect an expanding flow for forward motion: leftward flow on the left, rightward flow on the right. Also, it looks like the flow is a binary thing (left or right), while the EMD can estimate how quick the flow is. Can this not be better visualized?

4. Have I overlooked videos of the experiments? As a roboticist, videos are important, as they show properties that are not evident from a plot (e.g., how fast the motion is, etc.). Moreover, could the authors comment on whether their hardware setup allows for any logging? This in relation to the few failure cases.

5. Line 15-17: I would expect a bit more context in the introduction about work on optic-flow-based obstacle avoidance. Later, line 270 – 275, the authors comment on some other OF-based avoidance methods, but purely in the context of neuromorphic / non-neuromorphic. Work that I miss in the current article, but which is in my eyes very relevant:

a. Sanket, N. J., Singh, C. D., Ganguly, K., Fermüller, C., & Aloimonos, Y. (2018). Gapflyt: Active vision based minimalist structure-less gap detection for quadrotor flight. *IEEE Robotics and Automation Letters*, 3(4), 2799-2806.

b. de Croon, G. C., De Wagter, C., & Seidl, T. (2021). Enhancing optical-flow-based control by learning visual appearance cues for flying robots. *Nature Machine Intelligence*, 3(1), 33-41.

c. Hyslop, A. M., & Humbert, J. S. (2010). Autonomous navigation in three-dimensional urban environments using wide-field integration of optic flow. *Journal of guidance, control, and dynamics*, 33(1), 147-159.

6. “81 % of collision free travels. We examined the simulated robot’s performance to understand the essential behavioural components which led to such a low collision rate.” \diamond 19% collision rate does not sound very low. Perhaps the authors can give a bit more context here, why they consider this so low? The collisions depend on the clutter and conditions, so this could be part of the reflection. Another part could be commenting on how it compares to other robotics avoidance work.

Minor remarks:

1. “OF is the apparent motion of the surroundings on the animal’s retina when the animal translates within its environment”  OF can also be caused by rotations or by other objects moving. Suggestion: “moves relative to the environment”?

2. Fig. 3: The environment has high-contrast walls. It would be nice to have a reflection in the article how the approach generalizes to a normal environment. For example, is the event-camera good enough for that?

3. Line 150: "Our agent chooses a larger gap by using a simple probabilistic integration mechanism." It would be nice to have a brief explanation here how this works. In general, at multiple points in the text thresholds are mentioned which probably are not directly coded in the neurons, but depend on e.g., the weights of inhibitory and excitatory connections, etc. It would be good to explain how such things are tuned. (For example, the threshold on line 195)
4. Fig 4: "different environments"  "different simulated environments"?
5. The saccadic control strategy: I'm not sure that I really see the saccades in Fig 4 a-c. Perhaps zoom in once, or indicate with a star when there is a saccade happening?
6. Line 221: What are the authors referring to with the efference copy?
7. line 290: "for SNN simulation in operational real-time." \diamond implementation? Is it a simulation if in real hardware?
8. Line 338: "50% of its receptive field elicit 338 an event within a rolling window of 20ms." How is this implemented neurally?
9. Fig A3: There are 4 populations for the WTA. Can the robot only choose 4 directions?
10. Line 364: How did you tune it for 700 ms?
11. If MOT does not spike and it sends PWM, shouldn't the motors then be off / wheels not turning?
12. Line 400: what is the consequence of not having an OFI neuron?
13. How long do simulations last? Why only 3 per condition?

Reviewer #2 (Remarks to the Author):

This paper presents the use of artificial agents which allows for testing the functionality of the underlying neuronal processes in controlled environments. The authors explain aspects of the obstacle avoidance behavior of insects by modeling. A robotic system inspired by the behaviors and neurobiology of insects was developed. The model was implemented in event-based neuromorphic sensory processing hardware. The authors claimed that their neuromorphic system has capabilities for traveling in dense terrains, avoiding collisions, crossing narrow gaps, selecting safe passages, and maintaining a safe distance to objects. The claimed capabilities were illustrated by a neuromorphic network using action potentials to steer the robot toward regions of low apparent motion.

There is an attempt to develop a robotic system inspired by the behaviors and neurobiology of insects, however, the attempt lacks in-depth analyses and clear methodology. Please find my comments, questions, and concerns about the paper:

1-The overlap between this paper and previously presented work at the Nature Conference 2020 on Neuromorphic Computing in Beijing shall be clarified.

2-For event-based sampling, the signals are sampled when measurements pass a certain threshold, how these thresholds have been identified for different conditions? from my experience, those thresholds play a major role in the DVS camera output. For instance, if you make the thresholds small, the DVS camera noise will be significant, otherwise, you lose meaningful information. I could not find anywhere in the paper how did you solve this problem.

3-In the optical flow (OF) method for event cameras, did the authors combine those image-based methods with event data? If this is the case, it requires several adaptations (such as data conversion, and loss function) which have different properties. Otherwise, does the retinotopic map (OF) trained based on the contrast maximization framework to estimate optical flow from events alone? Did you investigate key elements for the OF in the Spiking Neural Network (SNN) model including how to design the objective function to prevent overfitting, how to warp events to deal better with occlusions, how to improve convergence with multi-scale raw events, how to deal with motion blur, surrounding brightness (including low visibility), occlusion and cue...etc.

4-Lines 24-35, the authors stated "we constrained our neuromorphic model agent by known properties of the fruit fly's visual motion path, for which the most is known among insects at the neural level". Where did you get the assumption from? it is not supported by any evidence or literature.

5-Lines 41-43, the authors stated "we mimicked the performance of T4/T5 cells, spatiotemporally integrated their responses, and processed these signals in a neural network providing direction information for locomotion control to avoid collisions". This is a major claim, but the methodology of developing this claim is not clear.

6-Lines 51-55, Spiking Neural Network (SNN) model composes of a retinotopic map, and an inverse soft Winner-Take-All (WTA) network was used. It is not clear to me if this structure was developed in this paper or borrowed from previous works. Why this particular structure was selected? There are no proper justifications apart from saying the retinotopic map does OF and WTA searches for a region of low apparent motion.

7-Lines 61-62, the authors stated "clearance of ~ 6 a.u. to objects in a box and to enter corridors only with a width greater than 10 a.u.", and the same unit was used for the velocities. The a.u. is a relative unit of measurement to a predetermined reference measurement, the predetermined reference measurement for space, positions, and velocities were not defined.

8-It is very hard to follow Figure 1, too much info. It could be divided into two figures to enhance the

clarity and readability of the figure.

9-Lines 75-77, The sEMD model is composed of two macro pixels (2×2 pixels), two Spatio-Temporal Correlation (SPTC) neurons, and one Time Difference Encoder. I assume that the spatial is x, and y, what about the depth information from an obstacle which is crucial for obstacle detection and avoidance? I reckon the agent is limited to 2D motion. In the robotics field, many advanced approaches were developed for such applications, Tesla autopilot is an example. The paper lacks novelty in this article and is neither theoretical nor practical.

10-Lines 90-91, the authors stated "This processing is part of the input to the flight control and obstacle avoidance machinery", the robotic system is a ground robot with slow motion, the validation could be more in-depth if an aerial robotic system was used. While in Section 3, the talk is about flying animals, flying insects require 3D perception.

11-Real experiments in "2.2.1 Corridors (Real World)" represent lab-controlled experiments (controlled environments). In this paper, what is called a cluttered environment is known in advance however the system shall deal with unknown cluttered environments including dynamic and unknown obstacles as insects do in real life.

12-The paper looks fragmented, it is a collection of works that are not well integrated. The paper in its current form does not have a significant novelty to the field.

13-Most of the experiments in this article were conducted in simulations, a very simple corridor-centering experiment was conducted in a lab environment (controlled environment). The viability of the presented work shall be tested using an aerial robotic system that requires 3D perception, at different speeds, and light intensities, in unknown environments with dynamic obstacles.

14-I could not access the two-web links provided in Sections 5 and 6.

15-I recommend providing a video to show the real experiments with overlaid text to illustrate the test protocol & scenarios, and the results.

16-Most of the paper is literature, the work does not include enough details on the methods in order to be reproduced or verified. The overall methodology is not sound and lacks in-depth analyses.

REVIEWER COMMENTS

Reviewer #1 (Remarks to the Author):

The authors propose a fully neuromorphic pipeline for optic-flow-based obstacle avoidance on a robot, with spikes propagating from an event camera through a spiking neural network and finally to motor actuators. The obstacle avoidance scheme is quite advanced, including slowing down when the environment gets cluttered, and performing navigation in saccades – similar to how flies navigate. The robot's "artificial brain" contains many elements that are non-standard in AI (besides the dynamics of spiking neurons, also inverse winner-take-all structures, etc.). The brain is completely handmade and tuned to exhibit desirable properties, such as passing through certain gap widths or turning away from them. These behaviors emerge from the neural parameters and dynamics. This is very interesting when compared to, e.g., common deep reinforcement learning practices.

In my opinion this work is very impressive and definitely forms the type of significant advance that Nature Communications intends to publish. Moreover, it is of relevance not only to robotics and engineering fields, but also to biology, as specific mechanisms are proposed that can potentially explain biological phenomena.

I do have a number of remarks that may be used to improve the manuscript.

Main remarks:

1. The advantages of neuromorphic pipelines would be energy efficiency and fast, asynchronous updates.
 - a. Can the energy expenditure of the neuromorphic pipeline be measured? How does it compare to, e.g., an Intel Realsense + NVidia Xavier? If measuring is not possible, can you estimate the energy costs? And what was the mass of all required components?

The energy efficiency is indeed an important measure. We estimate the maximum power since the pipeline is not up and running anymore. The processing pipeline consists of a DVS128PAER, AERnode board and a SpiN-3 board. SpiN-3 has 4 chips each of them consuming 250 up to 950mW [1]. The DVS128 chip consumes around 23mW [2]. The power consumption of the AERnode board varies depending on the application. However, we can assume that our implementation requires probably a comparable amount of power or even less than the implementation of another sensor into the AERnode board such as the implementation of the Neuromorphic Auditory Sensor by [3]. The Neuromorphic Auditory Sensor consumes around 30 mW. Hence, the whole computational system requires approximately 1-4 Watts.

We found another approach for obstacle avoidance using around 1040 neurons and approximately 8200 synapses [4]. This approach was successfully implemented on a Nvidia TX1 GPU which typically runs with 6-16 Watts [5]. Hence, the power consumption of our system is approximately 4 times lower. We added a sentence to the discussion, section 3.4, including this comparison.

The mass of the whole system including batteries, DVS128, AERnode board, robot chassis, motor controller and SpiN-3 board amounts to 2.3 kg. We added a sentence about this to the Figure of the robot.

1) I. Sugiarto, G. Liu, S. Davidson, L. A. Plana and S. B. Furber, "High performance computing on SpiNNaker neuromorphic platform: A case study for energy efficient image processing," *2016 IEEE 35th International Performance Computing and Communications Conference (IPCCC)*, Las Vegas, NV, USA, 2016, pp. 1-8, doi: 10.1109/IPCCC.2016.7820645.

2) P. Lichtsteiner, C. Posch and T. Delbruck, "A 128×128 120 dB 15μs Latency Asynchronous Temporal Contrast Vision Sensor," in *IEEE Journal of Solid-State Circuits*, vol. 43, no. 2, pp. 566-576, Feb. 2008, doi: 10.1109/JSSC.2007.914337.

3) A. Jiménez-Fernández *et al.*, "A Binaural Neuromorphic Auditory Sensor for FPGA: A Spike Signal Processing Approach," in *IEEE Transactions on Neural Networks and Learning Systems*, vol. 28, no. 4, pp. 804-818, April 2017, doi: 10.1109/TNNLS.2016.2583223.

4) Domcsek, Norbert & Knight, James & Nowotny, Thomas. (2018). Autonomous robot navigation using GPU enhanced neural networks. 77-79. 10.31256/UKRAS17.25.

5) Eisenbach, Markus & Stricker, Ronny & Seichter, Daniel & Vorndran, Alexander & Wengefeld, Tim & Gross, Horst-Michael. (2017). Speeding up Deep Neural Networks on the Jetson TX1.

b. The vision (optic flow) seems a bit "slow", tuned to 5Hz. But this may be an impression that is not true, since later (line 388) it is mentioned that the wave grating covers 20 degrees. Is 5 Hz then equal to 100 deg / second? In general, the authors use temporal frequency as a measure for optic flow speed, which is confusing in my eyes, because optic flow is an angular change over time – not a frequency. The authors would do well to clarify this matter.

In the literature on biological motion detectors, periodic patterns (gratings) were used for analysis. The tuning properties of motion detectors are calibrated on this basis. This is why temporal frequency is usually used to indicate velocity in these studies [1], [2], [3]. In this article we compare the response of our spiking elementary motion detector model to the response of T4/T5 neurons presented with such a periodic pattern (see Figure 2). This is why we chose to use gratings as visual input. You are right that a temporal frequency of 5 Hz in our experiment refers to the square wave grating moving with 100 degrees/second. In comparison the T4 and T5 neurons in walking *Drosophila* are tuned to a temporal frequency of 1 Hz [3]. Our robotic system is moving with approximately 15 cm/s, *Drosophila* walking speed is about 3cm/s. The frequency of the optic flow is proportional to the walking speed of the agent. Therefore, we think that this tuning is quite reasonable.

1) Madyam V. Srinivasan, *Honeybees as a Model for the Study of Visually Guided Flight, Navigation, and Biologically Inspired Robotics*, 2011

- 2) **Julien R. Serres, Franck Ruffier, Optic flow-based collision-free strategies: From insects to robots, 2017**
- 3) **Alexander Borst, Complementary mechanisms create direction selectivity in the fly, 2016**

2. The slowing down in cluttered environments is very interesting. Fig. 4e shows the result that goes with the finding that this helps to improve obstacle avoidance at high densities. However, this graph where a fixed velocity is better, between 10 and 20%. Moreover, whether one is better than the other in that graph is hard to evaluate, since there are no whiskers to indicate percentiles in the data. On how many runs are these results based? Is the reason for this that there are only three runs per condition? If so, would it be possible to do more runs?

The observation that the agent with a fixed velocity performs better between 10 and 20 percent is indeed very interesting. We think that the difference is probably insignificant. We repeated all experiments with a slightly different network as a response to your next comment. We can see that in the new runs this finding does not occur anymore. The drop in success rate from zero to 10 percent obstacle density is caused by a single crash of the robot so that we assume that this event is insignificant.

The run with changing velocity was performed 79 times, the run with fixed velocity 56 times. I added a sentence to Figure 4 containing this information. For each run a random number of obstacles and random obstacle positions were initialized.

The values displayed are absolute values. The number of successful collision free runs was counted and divided by the absolute number of runs. No statistics analysis can be performed on these data to estimate the standard deviation since no variation is given.

We increased the number of runs for all experiments to at least 10.

3. Visualization of the direction selection: Fig. 1 shows the winning direction with a purple bar. However, it would be interesting to see the INT layer as well, which (I guess) shows a more continuous picture of good and bad directions. Some more examples in the appendix would be appreciated. Fig 1 is very important, but in general could be improved: We see events at textureless locations (apparently these have grated patterns, but they are not shown). We do not see events at locations in the sky that have some cloud texture. The ground is textureless... I propose that the authors show texture on the walls. Moreover, the authors should comment on whether ground texture would interfere with their current scheme. I think it would, and although no showstopper (it can easily be ignored on a driving robot just in terms of image coordinates), it should be mentioned. Fig. 1c in my eyes almost shows random flow, whereas I would expect an expanding flow for forward motion: leftward flow

on the left, rightward flow on the right. Also, it looks like the flow is a binary thing (left or right), while the EMD can estimate how quick the flow is. Can this not be better visualized?

Thank you for your helpful feedback regarding Figure 1. The pattern in the sky is not received by the camera of the agent. We have added a sentence in the Figure legend

to clarify this. Furthermore, we have added texture to the object walls and have added a sentence to the figure caption explaining that ground texture would indeed be problematic if it is perceived by the agent and the optic flow pattern is later flattened to 1D.

After your comment on Figure 1c regarding the random flow we further analyzed the sensory output of the robot. One limitation of the sEMD is that it does not measure the optic flow accurately if the spatial frequency of the pattern is much higher than the time constant of the model. In that case, the model simply bypasses the signal without any direction sensitivity. Furthermore, the timing information provided by the simulated event based camera is not as accurate as the information of a real event based camera. After we received your feedback we implemented a new population between the SPTC population and TDE population to overcome these limitations. This layer acts as a kind of onset filter. Through self-inhibition it reduces the number of spikes, pronounces free standing edges and it reduces the retinotopic map from 2D to 1D. By reducing the frequency and pronouncing edges the TDEs are able again to detect the temporal component and the direction sensitivity of the optic flow. Hence, the network performs much more accurately than previously. The reduction from 2D to 1D provides sharper edges, hence more precise temporal information for the timing estimate. Furthermore, this reduces the number of neurons immensely in comparison to the old model.

With this new network model we repeated all experiments.

I tried to improve the visualization and the optic flow looks now as expected. We also added Figure A.10 to the appendix which shows the output of sEMD right - sEMD left for two short runs. Here we can see that the TDEs correctly detect the direction of OF at the edges. However, due to a lot of structure in the objects with a spatial frequency higher than the temporal frequency of the optic flow the optic flow direction detection is less precise inside the objects. This is not relevant for the obstacle avoidance task since the direction of OF does not play a role in that case. However, we will further investigate this limitation in the future. Some kind of shape detection, for example with gabor filters, before measuring the optic flow might be a solution to overcome this problem.

4. Have I overlooked videos of the experiments? As a roboticist, videos are important, as they show properties that are not evident from a plot (e.g., how fast the motion is, etc.). Moreover, could the authors comment on whether their hardware setup allows for any logging? This in relation to the few failure cases.

We have added an annotated video of the real world experiments to the new submission. We also added rasterplots of the hardware response to Figure 3 and Figure A12. These rasterplots show that the robot crashes in the tunnel when it cannot find an obstacle free direction for a long time period. In that case it just moves straight forward until it crashes.

5. Line 15-17: I would expect a bit more context in the introduction about work on optic-flow-based obstacle avoidance. Later, line 270 – 275, the authors comment on some

other OF-based avoidance methods, but purely in the context of neuromorphic / non-neuromorphic. Work that I miss in the current article, but which is in my eyes very relevant:

Thank you for suggesting the following article. In the former line 15-17 we restrained the mention to methods closely related to observed behaviour of insects be they flying in tunnel or around obstacles. We now included de Croon et al and Hyslop et al in the introduction as well. However, Sanket et al is a bit further away from the insect inspired literature, as they rely on deep-learning method to map the optic-flow to a gap. According to the literature on insects, the approach by de Croon based on self-learning is much closer to observed behaviour in insects. Nevertheless we mentioned Sanket et al approach as well as Hyslop et al. in the Table 1.

- a. **Sanket, N. J., Singh, C. D., Ganguly, K., Fermüller, C., & Aloimonos, Y. (2018). Gapflyt: Active vision based minimalist structure-less gap detection for quadrotor flight. IEEE Robotics and Automation Letters, 3(4), 2799-2806.**
- b. **de Croon, G. C., De Wagter, C., & Seidl, T. (2021). Enhancing optical-flow-based control by learning visual appearance cues for flying robots. Nature Machine Intelligence, 3(1), 33-41.**
- c. **Hyslop, A. M., & Humbert, J. S. (2010). Autonomous navigation in three-dimensional urban environments using wide-field integration of optic flow. Journal of guidance, control, and dynamics, 33(1), 147-159.**

6. “81 % of collision free travels. We examined the simulated robot’s performance to understand the essential behavioural components which led to such a low collision rate.” 19% collision rate does not sound very low. Perhaps the authors can give a bit more context here, why they consider this so low? The collisions depend on the clutter and conditions, so this could be part of the reflection. Another part could be commenting on how it compares to other robotics avoidance work.

81 percent seems indeed not very high at a first glance. After repeating all the experiments with the new architecture the mean success rate of our system lies at 97 percent. We added Table 1 to the paper to compare this performance to other approaches. Indeed, with 90-100 percent of success our approach is comparable to other state of the art methods. We added a sentence to the end of the discussion pointing out to the table.

Minor remarks:

1. “OF is the apparent motion of the surroundings on the animal’s retina when the animal translates within its environment”  OF can also be caused by rotations or by other objects moving. Suggestion: “moves relative to the environment”?

Thank you for this suggestion. We added translational to the beginning of the sentence since here we want to especially refer to translational motion.

2. Fig. 3: The environment has high-contrast walls. It would be nice to have a reflection in the article how the approach generalizes to a normal environment. For example, is the event-camera good enough for that?

We added some additional text to the last section of the discussion explaining the challenges of natural environments:

Natural environments lead to new challenging scenarios such as low contrast walls which cannot be detected by event-based cameras. The optic flow of such areas can be predicted by interpolating the shape and area of the object from the events at the edges. Another option would be to add a different type of sensor, such as radar, which are able to detect structureless objects. By extending our work in these directions we can make it even more robust for real world examples.

3. Line 150: “Our agent chooses a larger gap by using a simple probabilistic integration mechanism.” It would be nice to have a brief explanation here how this works. In general, at multiple points in the text thresholds are mentioned which probably are not directly coded in the neurons, but depend on e.g., the weights of inhibitory and excitatory connections, etc. It would be good to explain how such things are tuned. (For example, the threshold on line 195)

In this case we are referring to the mechanism described three sentences later. We added a sub sentence referring to the later text and also referring to the description of the obstacle avoidance network in the methods section.

4. Fig 4: “different environments”  “different simulated environments”?

We added “simulated” to the text.

5. The saccadic control strategy: I’m not sure that I really see the saccades in Fig 4 a-c. Perhaps zoom in once, or indicate with a star when there is a saccade happening?

It is indeed impossible to see the saccadic turns in these plots. Hence, we included a new Figure A.10 in the appendix in which the turns are clearly visible. We also refer to it in the caption of Figure 4.

6. Line 221: What are the authors referring to with the efference copy?

In our article the efference copy is the inhibitory feedback from the motor populations to the inverse WTA and to the SPTC layer. We added this information to the discussion.

7. line 290: “for SNN simulation in operational real-time.” implementation? Is it a simulation if in real hardware?

This is indeed a good question. The SpiNNaker board consists of a number of ARM cores, each of them contains up to 250 neurons. These neuron models are written in c

code and are not physically implemented on the hardware, they are not emulated. Therefore, I would personally call it a simulation of the neurons.

8. Line 338: "50% of its receptive field elicit 338 an event within a rolling window of 20ms." How is this implemented neurally?

The spiking threshold of the SPTCs is set very high so that at minimum two consecutive events in a short time window are required to increase the membrane potential enough to overcome the threshold. If the time between the two spikes is too long, the membrane potential decreases back to its resting state before it can reach the threshold. We added a few sentences explaining this to the section Obstacle Avoidance Network.

9. Fig A3: There are 4 populations for the WTA. Can the robot only choose 4 directions?

The WTA contains four subpopulations. The 18 neurons in the center all represent different movement directions. All neurons in the subpopulations on the left and right have the same movement duration, to the left and right respectively. Hence, there are 20 different movement directions. We added a sentence about this to the Figure caption.

10. Line 364: How did you tune it for 700 ms?

The 700 ms are caused by a complex interplay between Poisson input spike trains, motor inhibition and optic flow inhibition. This tuning process requires a lot of experience. By increasing the Poisson spike train rate the number of decisions can be increased. However, a too high number can lead to faulty spikes. The inhibition from the motor population plays a crucial role. If it is too strong a new decision takes forever, if it is too weak a new decision will be taken before the turn is over. This is a very interesting question but I think this information is too detailed for the manuscript so that I would not include it.

11. If MOT does not spike and it sends PWM, shouldn't the motors then be off / wheels not turning?

MOT sets the PWM signal in the FPGA, however, the FPGA continues sending the last PWM signal as long as no update occurs. We added a sentence regarding this to the Obstacle Avoidance Network section.

12. Line 400: what is the consequence of not having an OFI neuron?

The OFI neuron is used to modify the velocity of the agent. Since the real-world agent moves with constant velocity no OFI neuron is required. We rephrased this in the article.

13. How long do simulations last? Why only 3 per condition?

When repeating the experiments we increased the number of all simulations to at least 10. Simulations of the cluttered environment last 4 hours, other simulations 2 hours. We added this information to the methods section Neurorobotics Platform (Simulation).

Reviewer #2 (Remarks to the Author):

This paper presents the use of artificial agents which allows for testing the functionality of the underlying neuronal processes in controlled environments. The authors explain aspects of the obstacle avoidance behavior of insects by modeling. A robotic system inspired by the behaviors and neurobiology of insects was developed. The model was implemented in event-based neuromorphic sensory processing hardware. The authors claimed that their neuromorphic system has capabilities for traveling in dense terrains, avoiding collisions, crossing narrow gaps, selecting safe passages, and maintaining a safe distance to objects. The claimed capabilities were illustrated by a neuromorphic network using action potentials to steer the robot toward regions of low apparent motion.

There is an attempt to develop a robotic system inspired by the behaviors and neurobiology of insects, however, the attempt lacks in-depth analyses and clear methodology. Please find my comments, questions, and concerns about the paper:

1-The overlap between this paper and previously presented work at the Nature Conference 2020 on Neuromorphic Computing in Beijing shall be clarified.

Most of this work was presented at the Neuromorphic Computing Conference 2020 in Beijing. However, this was a purely internal conference with no conference proceedings. According to the guidelines at <https://www.nature.com/ncomms/editorial-policies/press-and-embargo-policies> conference presentations do not constitute prior publication.

2-For event-based sampling, the signals are sampled when measurements pass a certain threshold, how these thresholds have been identified for different conditions? from my experience, those thresholds play a major role in the DVS camera output. For instance, if you make the thresholds small, the DVS camera noise will be significant, otherwise, you lose meaningful information. I could not find anywhere in the paper how did you solve this problem.

We used the slow dvs128 bias setting provided in the jAER framework for the sEMD characterization and real world experiments. In the sEMD characterization experiment we saw that the sEMDs together with the DVS128 and the dvs128 bias setting provide a performance comparable to the fruit fly. Hence, we concluded that this bias setting is well suited to compare the performance of the robot to fruit flies and other insects. We added a new sentence with the used bias setting to section 2.1. By integrating the events from 2x2 pixels onto a single LIF neuron called SPTC layer, which does not show any response to single noisy events, we filtered out a great

amount of noise. We added a sentence to section 2.1. explaining this. This is also mentioned in the methods in section 4.3. line 338ff.

3-In the optical flow (OF) method for event cameras, did the authors combine those image-based methods with event data? If this is the case, it requires several adaptations (such as data conversion, and loss function) which have different properties. Otherwise, does the retinotopic map (OF) trained based on the contrast maximization framework to estimate optical flow from events alone? Did you investigate key elements for the OF in the Spiking Neural Network (SNN) model including how to design the objective function to prevent overfitting, how to warp events to deal better with occlusions, how to improve convergence with multi-scale raw events, how to deal with motion blur, surrounding brightness (including low visibility), occlusion and cue...etc.

This work does not use any machine learning methods, any deep learning, or any learning in general. Furthermore, we do not use any frame-based methods. We use the DVS128 which does not comprise any frame based sensor together with spiking neural networks. The different layers of the network are generated and tuned by hand once and kept constant for all experiments. No contrast maximization, training data or objective function are given. Our scope is not to outperform machine learning based obstacle avoidance methods, but explains underlying biological processes with a fully close-loop system and provides new methods on how to build spiking neural networks by hand, closely inspired by insects.

Most of the problems you are mentioning are irrelevant for obstacle avoidance using the sEMD. Our robotic agent has to avoid close by objects by finding gaps. Since the robot does not have a goal direction it does not care about further away objects that might be occluded by close by objects. There is no convergence since there is no learning. Motion blur is optic flow, this is the cue used to estimate distance. We prove in the sEMD characterization that the sEMD is very robust against changes in illumination (See appendix Figure A2.a). The model also works well for relative contrasts between 40% and 100% (See appendix Figure A2.b). As expected, patterns with very low contrast are difficult to detect.

The main components of this paper are the spiking elementary motion detector (sEMD) and the inverted winner-take-all. The elementary motion detector converts temporal information such as optic flow, the time to travel of an edge from one pixel to another, into a number of events in a direction-sensitive manner.

While a deep neural network for optic flow estimation needs to be trained the sEMD provides optic flow as an output by its nature. This makes the sEMD a compact low-level-model for optical flow estimation. A more detailed explanation of the sEMD is given in [1]. The optic flow is passed on to an inverse winner-take-all that extracts a direction of low apparent motion. I hope this clarifies our methodology.

- 1) Milde MB, Bertrand OJN, Ramachandran H, Egelhaaf M, Chicca E. Spiking Elementary Motion Detector in Neuromorphic Systems. Neural Comput. 2018 Sep;30(9):2384-2417. doi: 10.1162/neco_a_01112. Epub 2018 Jul 18. PMID: 30021082.**

4-Lines 24-35, the authors stated “we constrained our neuromorphic model agent by known properties of the fruit fly’s visual motion path, for which the most is known among insects at the neural level”. Where did you get the assumption from? it is not supported by any evidence or literature.

The processing of motion vision in insects has been studied on different insects, such as beetles, bees, and flies. Thanks to the advance of genetic tools in drosophila, the connectomics of different neurons can be established, and the neuronal response of small neurons (such as the T4/T5 cells) can be obtained. Therefore, the understanding at a neuronal level of motion vision processing in fruit flies is better understood than in the other species. Although several actors in the field would agree about our statement, we are unaware of a systematic review and meta-analyses comparing the knowledge of the motion vision system of different insect species. We, therefore, rephrase the sentence and source it to a review on optic flow-based course control in insects.

5-Lines 41-43, the authors stated “we mimicked the performance of T4/T5 cells, spatiotemporally integrated their responses, and processed these signals in a neural network providing direction information for locomotion control to avoid collisions”. This is a major claim, but the methodology of developing this claim is not clear.

The tuning curve of T4 and T5 cells relative to the temporal frequency of the square wave grating is bell shaped. We quantify the sEMD tuning curve employing the same method as for T4 and T5 curve as described briefly in the result section and in more detailed in the Appendix A.1 and A.2. The term “performance” is not fitting our claim, because we did not define the performance of T4 and T5 cells, which in biological term will be difficult to define. We therefore replaced “performance” by “neuronal response”.

6-Lines 51-55, Spiking Neural Network (SNN) model composes of a retinotopic map, and an inverse soft Winner-Take-All (WTA) network was used. It is not clear to me if this structure was developed in this paper or borrowed from previous works. Why this particular structure was selected? There are no proper justifications apart from saying the retinotopic map does OF and WTA searches for a region of low apparent motion.

The motivation of a retinotopic organization of the optic-flow response is inspired by the retinotopically organized T4 and T5 cell in a fruit fly brain. These cells on the one hand project to large neurons (tangential cells) but also likely project to the central part of the brain, as this part is also sensitive to optic-flow, and a center of integration of different sensory modalities. The inverse soft WTA provides the advantage in contrast to balancing the optic-flow or integrating it on the entire visual field, to search for regions of low apparent motion, and thus might be more robust in a variety of environments (as we have shown in our manuscript). In addition this network was suggested in previous studies, but only under open-loop conditions. The usability of such an approach in close-loop, as experienced by the insects, were therefore

unknown before our study. We have clarified the underlying motivation for the architecture.

7-Lines 61-62, the authors stated “clearance of ~6 a.u. to objects in a box and to enter corridors only with a width greater than 10 a.u.”, and the same unit was used for the velocities. The a.u. is a relative unit of measurement to a predetermined reference measurement, the predetermined reference measurement for space, positions, and velocities were not defined.

The unit is defined in the method section line 462f. We added it to the main text now so that it is easier to find.

We decided to use this relative unit to make the experiments more comparable to insects, since they are operating on a different spatial scale.

8-It is very hard to follow Figure 1, too much info. It could be divided into two figures to enhance the clarity and readability of the figure.

All subplots in Figure 1 belong together and explain the structure and main principle of the obstacle avoidance network. We tried to further simplify the readability by adding labels to subplot c-e.

9-Lines 75-77, The sEMD model is composed of two macro pixels (2×2 pixels), two Spatio-Temporal Correlation (SPTC) neurons, and one Time Difference Encoder. I assume that the spatial is x, and y, what about the depth information from an obstacle which is crucial for obstacle detection and avoidance? I reckon the agent is limited to 2D motion. In the robotics field, many advanced approaches were developed for such applications, Tesla autopilot is an example. The paper lacks novelty in this article and is neither theoretical nor practical.

Here the reviewer focuses on applications of obstacle avoidance for drones and autonomous driving. The mentioned approaches work for sure very well, however, they are not designed to explain biology. Our paper provides a new point of view on how obstacle avoidance could be performed by insects. We use a strongly bio-inspired approach and show that it can perform obstacle avoidance, tunnel centering and gap choice. Most other insect-inspired models can only explain one mechanism such as optical flow balancing for tunnel centering or keeping the optical flow constant for smooth landing. Thus, we provide a new hypothesis on how obstacle avoidance, gap choice and tunnel centering could happen in the fly brain. Flying insects only use optical flow from 2D motion to estimate distance. They are very good at performing obstacle avoidance with a power consumption in the range of milliwatts and a compact, low-resolution neural system for navigational tasks. This paper gives a new perspective on how obstacle avoidance and other navigational tasks could be performed with very limited sensory and computational resources. Using only one camera instead of two cameras or a lidar keeps the power consumption, computational complexity and resources required on a low level. In the

long run this work might lead to the development of power-efficient and robust obstacle avoidance using networks inspired by the connectomics, electrophysiological recordings and behavioral findings in insects. This approach might provide an alternative for machine learning methods and deep neural networks. The paper serves as a new perspective on how insects perform these tasks.

10-Lines 90-91, the authors stated "This processing is part of the input to the flight control and obstacle avoidance machinery", the robotic system is a ground robot with slow motion, the validation could be more in-depth if an aerial robotic system was used. While in Section 3, the talk is about flying animals, flying insects require 3D perception.

We repeated all experiments based on the comments of reviewer 1. Now the robot moves with a velocity of around 4 robot lengths (4 x 0.3m) per second. For a driving car that would be equivalent to a speed of approximately 10 meters per second which is 36 km/h. Most observations regarding flying insect behavior were performed in the plane. Very little is known about vertical flight behavior in flies and bees. Hence, moving the experiments to a drone setup creates a large overhead while changing very little regarding the explored computational primitives. The complexity would just be increased by one dimension and a number of new problems would occur which do not add anything new to the algorithm.

11-Real experiments in "2.2.1 Corridors (Real World)" represent lab-controlled experiments (controlled environments). In this paper, what is called a cluttered environment is known in advance however the system shall deal with unknown cluttered environments including dynamic and unknown obstacles as insects do in real life.

The cluttered environments are not known by the agent. The agent was tuned according to the experiments in the appendix and then applied to the different tasks. Since the robot does not involve any learning it does not generate a knowledge of the environment.

In the cluttered environment, all objects are placed at random positions. Objects are 1x1x2 meter blocks with a grating pattern. Since the objects are placed randomly in more than 50 runs all possible combinations of object positions and object clustering occur. The clustering of objects in the arena leads to new random shapes, such as walls, gaps, corridors and larger objects. The only factor that is kept constant is the pattern on the objects. However, we show in the sEMD characterization experiment for which range of contrasts the optic flow extraction performs well (see Figure A2b). Hence, we can argue that the obstacle avoidance network performs well in unknown random environments as long as the patterns on the obstacles have a contrast higher than 40 percent.

Nothing is known about how insects avoid dynamic obstacles. That could indeed be a nice future extension of the provided work. However, it requires prediction of the object's future position which is not provided in this work and not the aim of this work.

12-The paper looks fragmented, it is a collection of works that are not well integrated. The paper in its current form does not have a significant novelty to the field.

We rephrase the paper according to the comments. We hope this provides a clearer line of arguments. Overall the paper evaluates how a possible obstacle avoidance approach in insects performs on experiments typically done with insects.

13-Most of the experiments in this article were conducted in simulations, a very simple corridor-centering experiment was conducted in a lab environment (controlled environment). The viability of the presented work shall be tested using an aerial robotic system that requires 3D perception, at different speeds, and light intensities, in unknown environments with dynamic obstacles.

Thank you for suggesting a different way to perform experiments with the developed network. As mentioned above an aerial robotic system unnecessarily increases the computational overhead without adding new insights. Absolute depth perception is not required since we use optical flow for depth perception. Two different speed modes have already been tested, fixed and adaptive velocity. Different contrasts, velocities and illuminations were characterized in the sEMD evaluation experiment. All environments were unknown to the agent. This work aims at explaining how obstacle avoidance is performed in insects. Obstacle avoidance of dynamic objects was not the focus of this paper and would require many more computational modules to perform for example prediction of the object's future location. However, this is not the goal of this paper.

14-I could not access the two-web links provided in Sections 5 and 6.

The code is available here:

https://github.com/thorschoepe/collision_avoidance_SpiNNaker

https://github.com/thorschoepe/collisionavoidance_NRP

15-I recommend providing a video to show the real experiments with overlaid text to illustrate the test protocol & scenarios, and the results.

We added videos to the submission.

16-Most of the paper is literature, the work does not include enough details on the methods in order to be reproduced or verified. The overall methodology is not sound and lacks in-depth analyses.

REVIEWERS' COMMENTS

Reviewer #1 (Remarks to the Author):

I thank and congratulate the authors for making such a thorough revision, further improving their approach and reperforming all experiments. The new success percentage (90-100%) is indeed really good, and is now compared to other relevant results from the literature. I am satisfied by all responses to my comments, and in my opinion the article can now be accepted.

There are two small points for possible improvement that I will leave to the authors' discretion:

1. I like the video, as it shows the real-world experiments. The authors could also make an overview video explaining the concept and showing also some of the simulation experiments. It would be nice to see, for instance, how the simulated agent slows down when there is a lot of clutter in the environment.
2. On the expression of optical flow speed in Hz, perhaps add on line 79: (100 deg/s) after the "5 Hz".

Finally, congratulations on this very interesting work!

Reviewer #2 (Remarks to the Author):

The authors addressed my comments to a satisfactory level.

REVIEWERS' COMMENTS

Reviewer #1 (Remarks to the Author):

I thank and congratulate the authors for making such a thorough revision, further improving their approach and reperforming all experiments. The new success percentage (90-100%) is indeed really good, and is now compared to other relevant results from the literature. I am satisfied by all responses to my comments, and in my opinion the article can now be accepted.

There are two small points for possible improvement that I will leave to the authors' discretion:

1. I like the video, as it shows the real-world experiments. The authors could also make an overview video explaining the concept and showing also some of the simulation experiments. It would be nice to see, for instance, how the simulated agent slows down when there is a lot of clutter in the environment.

Thank you for your positive feedback. We generated an overview video which explains the neuromorphic algorithm and shows some results of the real world and simulation experiments. This video replaces the old video since the old video is completely included in this one as well.

2. On the expression of optical flow speed in Hz, perhaps add on line 79: (100 deg/s) after the "5 Hz".

We added 100 deg/s in line 79.

Finally, congratulations on this very interesting work!

Reviewer #2 (Remarks to the Author):

The authors addressed my comments to a satisfactory level.

Thank you for your thorough review.